# Androgen deprivation upregulates SPINK1 expression and potentiates cellular plasticity in prostate cancer

Ritika Tiwari[1,10], Nishat Manzar [1,10], Vipul Bhatia [1], Anjali Yadav[1], Mushtaq A. Nengroo[2], Dipak Datta[2], Shannon Carskadon[3], Nilesh Gupta[4], Michael Sigouros [5], Francesca Khani [6], Matti Poutanen[7], Amina Zoubeidi[8], Himisha Beltran[9], Nallasivam Palanisamy[3] & Bushra Ateeq [1]*

Emergence of an aggressive androgen receptor (AR)-independent neuroendocrine prostate cancer (NEPC) after androgen-deprivation therapy (ADT) is well-known. Nevertheless, the majority of advanced-stage prostate cancer patients, including those with SPINK1-positive subtype, are treated with AR-antagonists. Here, we show AR and its corepressor, REST, function as transcriptional-repressors of *SPINK1*, and AR-antagonists alleviate this repression leading to *SPINK1* upregulation. Increased SOX2 expression during NE-transdifferentiation transactivates *SPINK1*, a critical-player for maintenance of NE-phenotype. SPINK1 elicits epithelial-mesenchymal-transition, stemness and cellular-plasticity. Conversely, pharmacological Casein Kinase-1 inhibition stabilizes REST, which in cooperation with AR causes *SPINK1* transcriptional-repression and impedes SPINK1-mediated oncogenesis. Elevated levels of SPINK1 and NEPC markers are observed in the tumors of AR-antagonists treated mice, and in a subset of NEPC patients, implicating a plausible role of SPINK1 in treatment-related NEPC. Collectively, our findings provide an explanation for the paradoxical clinical-outcomes after ADT, possibly due to SPINK1 upregulation, and offers a strategy for adjuvant therapies.

[1] Molecular Oncology Laboratory, Department of Biological Sciences and Bioengineering, Indian Institute of Technology Kanpur, Kanpur, UP 208016, India. [2] Division of Cancer Biology, CSIR-Central Drug Research Institute, Lucknow, UP 226031, India. [3] Vattikuti Urology Institute, Department of Urology, Henry Ford Health System, Detroit, MI 48202, USA. [4] Department of Pathology, Henry Ford Health System, Detroit, MI 48202, USA. [5] Division of Medical Oncology, Weill Cornell Medicine, New York, NY 10065, USA. [6] Department of Pathology and Laboratory Medicine, Weill Cornell Medicine, New York, NY 10065, USA. [7] Institute of Biomedicine, Research Centre for Integrative Physiology and Pharmacology, University of Turku, Turku, Finland. [8] Vancouver Prostate Centre and Department of Urologic Sciences, University of British Columbia, Vancouver, BC V6T 1Z4, Canada. [9] Department of Medical Oncology, Dana Farber Cancer Institute, Harvard Medical School, Boston, MA 02215, USA. [10] These authors contributed equally: Ritika Tiwari, Nishat Manzar. *email: bushra@iitk.ac.in

Genetic rearrangement involving androgen-driven promoter of the serine protease, *TMPRSS2* and the coding region of *ERG*, a member of *ETS* (E26 transformation-specific) transcription factor family represents half of the prostate cancer (PCa) cases[1]. Subsequently, fusion involving other *ETS* family members (*ETV1*, *FLI1,* and *NDRG1*); *RAF* kinase rearrangements; *SPOP/CHD1* alterations; mutations in *FOXA1* and *IDH1* have also been discovered[2–4]. Overexpression of SPINK1 (Serine Peptidase Inhibitor, Kazal type 1) constitutes a substantial ~10–25% of the total PCa cases exclusively in *ETS*-fusion negative subtype[5,6]. Moreover, the expression of SPINK1 and ERG were shown in two distinct foci within a prostate gland, indicating that these two events are either independent or SPINK1 overexpression to be a sub-clonal event after *TMPRSS2-ERG* fusion[7]. Notably, SPINK1-positive patients show rapid progression to castration resistance and biochemical recurrence compared to *ETS*-fusion positive cases[5,8,9]. SPINK1, also known as tumor-associated trypsin inhibitor (TATI) or pancreatic secretory trypsin inhibitor (PSTI) was previously discovered in the urine of ovarian cancer patients[10]. Under normal physiological condition, SPINK1 inhibits the premature activation of pancreatic proteases, however, multiple reports have observed elevated levels of SPINK1 in cancer tissues, and shown its role in cancer progression[11–14]. Moreover, SPINK1 acts as an autocrine/paracrine factor and imparts oncogenic traits via EGFR downstream signaling[11,15].

Androgen deprivation therapy (ADT) remains the gold-standard for treating advanced PCa, however the disease often progresses as castrate-resistant prostate cancer (CRPC), associated with poor prognosis[16,17]. Sustained androgen signaling in CRPC tumors has been reported via multiple alteration in the *AR* gene or AR-signaling pathway such as mutations in its ligand binding domain (F877L and T878A), constitutively active variants (AR-V7 and ARv567es), amplification, or activation of AR-targets through steroid-inducible glucocorticoid receptor[18–20]. Current treatment regimen for CRPC patients include enzalutamide (MDV3100) and apalutamide (ARN-509) (which blocks AR nuclear translocation and its genomic binding), and abiraterone acetate (an irreversible steroidal CYP17A1 inhibitor, that targets adrenal and intratumoral androgen biosynthesis)[21–23]. Although, these AR-targeted therapies are known to prolong the overall survival of patients, the response is temporary, and the disease eventually progresses. A subset of CRPC patients (~20% of advanced drug-resistant cases) escape the selective pressure of AR-targeted therapies by minimizing the dependency on AR signaling and often through lineage plasticity and acquisition of a neuroendocrine PCa (NEPC) phenotype. Treatment-related NEPC is associated with poor prognosis and patient outcome[24]. NEPC exhibits a distinct phenotype characterized by reduced or no expression of AR and AR-regulated genes, and increased expression of NEPC markers such as synaptophysin (SYP), chromogranin A (CHGA), and enolase 2 (ENO2)[25]. Several molecular mechanisms have been proposed for CRPC to NEPC progression, including, frequent genomic alterations in *TP53* (tumor protein p53) and *RB1* (retinoblastoma-1-encoding gene)[26,27]. Moreover, *MYCN* amplification, BRN2 upregulation, mitotic deregulation via Aurora kinase A (AURKA), alternative splicing by serine/arginine repetitive matrix4 (SRRM4), and loss of repressor element-1 (RE-1) silencing transcription factor (REST), a transcriptional co-regulator of AR, are known to have a role in NE transdifferentiation[28–31].

Although, overexpression of SPINK1, which is seen in ~10–25% of PCa patients, has been associated with adverse clinical outcomes, the regulatory mechanism and the functional significance of SPINK1 upregulation remains largely unexplored. In this study, we discover that *SPINK1* is transcriptionally repressed by the AR and its co-repressor REST, and

AR-antagonists relieve this repression leading to SPINK1 upregulation. Moreover, we identify that reprogramming factor SOX2 positively regulates *SPINK1* during NE-transdifferentiation. Notably, we also show elevated SPINK1 levels in androgen-signaling ablated mice xenograft models and NEPC patients, highlighting its possible role in cellular plasticity and development of the NEPC phenotype. Collectively, our findings draw attention towards the widespread use of AR antagonists and the plausible emergence of a distinct resistance mechanism associated with ADT-induced SPINK1 upregulation in prostate cancer.

## Results

**SPINK1 and AR are inversely correlated in PCa patients.** Altered AR signaling and AR-binding have been studied extensively in localized PCa and CRPC[32]. It has been shown that AR binds with other cofactors, such as GATA2, octamer transcription factor 1 (Oct1), Forkhead box A1 (FoxA1) and nuclear factor 1 (NF-1) to mediate cooperative transcriptional activity of AR target genes[33]. Thus, we sought to discover the possible link between *SPINK1* and *AR* expression in PCa patients, and stratified patients available at TCGA-PRAD (The Cancer Genome Atlas Prostate Adenocarcinoma) cohort based on high and low expression of *AR*. The patients with higher expression of *AR* showed a significantly lower expression of *SPINK1* and contrariwise (Fig. 1a). To further confirm this association, we performed immunohistochemical (IHC) analysis for the expression of SPINK1 and AR on tissue microarrays (TMA) comprising PCa patient specimens ($n = 237$). Important to note that all of these cases underwent radical prostatectomy without any hormone or radiation therapy. In concordance with TCGA data analysis, our IHC findings reveal that SPINK1-positive patients exhibit low or negative staining for AR expression, while SPINK1-negative patients show high or medium AR staining (Fig. 1b and Supplementary Fig. 1a). Importantly, about ~67% of the SPINK1-positive patients (34 out of 51) demonstrate either low or negative staining for AR expression (Fisher's exact test, $P = 0.0004$) (Fig. 1c, d). Based on our findings, we conjecture that *SPINK1* is one of the AR repressed genes, hence we next examined the expression of AR and other members of AR repressor complex (*NCOR1*, *NCOR2,* and *NRIP1*) using TCGA-PRAD cohort, and the patients were sorted based on *SPINK1* high and low expression by employing quartile-based normalization[34]. Interestingly, we found that *SPINK1* expression is also negatively associated with other AR repressive complex members (Supplementary Fig. 1b). In addition, we investigated the correlation of *SPINK1* and AR signaling score using transcriptomic data from two independent PCa cohorts, Memorial Sloan Kettering Cancer Center (MSKCC) and TCGA-PRAD. As expected, a lower AR signaling score in *SPINK1*-positive patients was recorded compared to the *SPINK1*-negative patients (Supplementary Fig. 1c). Taken together, our findings show an inverse association between *SPINK1* expression and AR signaling in PCa patients, indicating that upregulation of SPINK1 is owing to the loss of AR-mediated repression during PCa progression.

**AR antagonists trigger SPINK1 upregulation in PCa.** Since an inverse association between *SPINK1* expression and AR signaling was observed in three independent PCa cohorts (TCGA-PRAD, MSKCC and ours) (Fig. 1), we examined role of AR signaling in the regulation of SPINK1 using PCa cell lines, 22RV1 (endogenously SPINK1-positive) and androgen responsive VCaP cells (*TMPRSS2-ERG* fusion positive) (Supplementary Fig. 2a, b). Stimulating 22RV1 cells with synthetic androgen, R1881 (10 nM), results in a significant decrease in expression of SPINK1 with a concomitant increase in the expression of AR target gene, *KLK3*

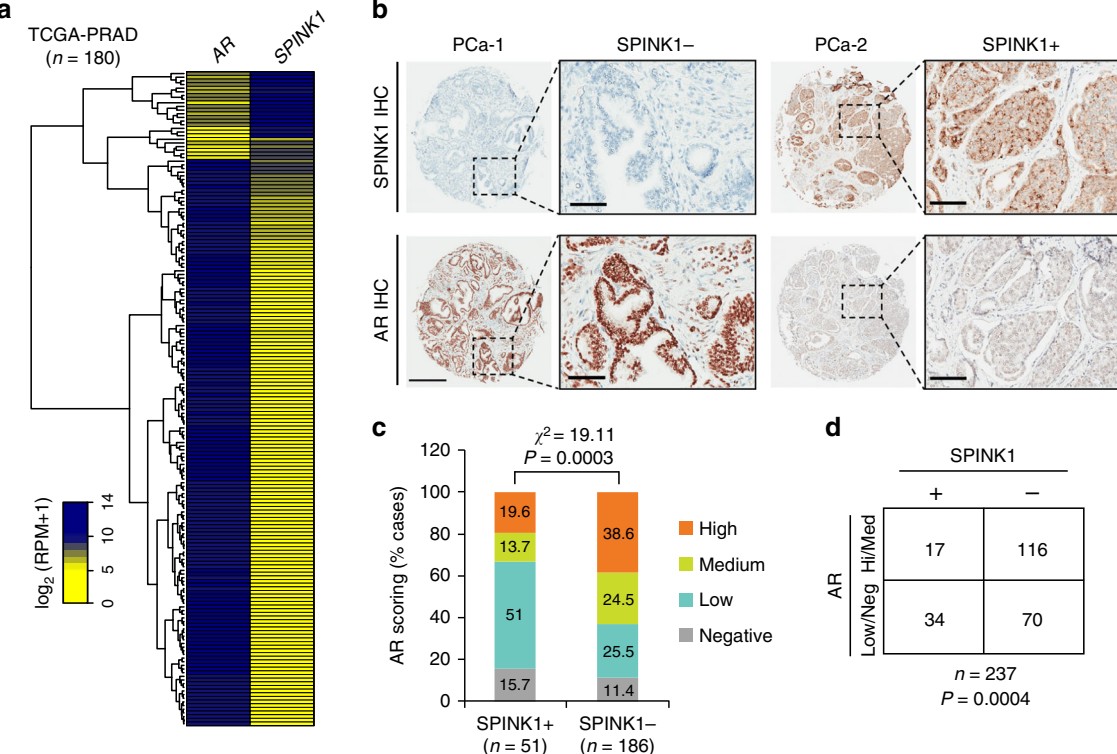

**Fig. 1 SPINK1 is negatively correlated with AR in PCa patients. a** Heatmap depicting *AR* and *SPINK1* expression in TCGA-PRAD cohort ($n = 180$). Shades of yellow and blue represents expression values in log$_2$ (RPM+1). **b** Representative micrographs depicting PCa tissue microarray (TMA) cores ($n = 237$), immunostained for SPINK1 and AR expression by immunohistochemistry (IHC). Top panel shows representative IHC for SPINK1 in SPINK1-negative (SPINK1−) and SPINK1-positive (SPINK1+) patients. Bottom panel represents IHC for AR expression in the tumor core from same patients. Scale bar represents 500 μm and 100 μm for the entire core and the inset images, respectively. **c** Bar plot showing percentage of IHC scoring for AR in the SPINK1+ and SPINK1− patients' specimens. *P*-value for the Chi-Square test is indicated. **d** Contingency table for the AR and SPINK1 status. Patients showing high or medium expression of AR were grouped as AR-(Hi/Med), while patients with low or null AR expression were indicated as AR-(Low/Neg). *P*-value for Fisher's exact test is indicated.

(Fig. 2a–c). A panel of SPINK1 positive and negative cancer cell lines were used to confirm the specificity of the SPINK1 antibody by immunostaining (Supplementary Fig. 2c). To further investigate whether similar effect on SPINK1 expression could be rendered by sub-physiological concentration of androgen, 22RV1 cells were stimulated with much lower concentrations of R1881 (0.01 and 0.1 nM), and interestingly both ~0.1 nM and 1 nM of R1881 were equally efficacious in repressing the expression of *SPINK1* transcript (Supplementary Fig. 2d). Similarly, VCaP cells stimulated with R1881 (10 nM) show a significant decline in the expression of SPINK1 both at transcript and protein levels, while an increase in the expression of *KLK3* was noticed (Fig. 2d–f). A remarkable decrease in the *SPINK1* expression was also noted even at sub-physiological concentration of androgen in VCaP cells (Supplementary Fig. 2e). We also analyzed the publicly available datasets (GSE71797 and GSE51872), wherein 22RV1 and VCaP cells were stimulated with R1881 and dihydrotestosterone (DHT), respectively, which exhibits reduced expression of *SPINK1*, among the several previously known AR repressed genes, namely *DDC*, *OPRK1*, *NOV,* and *SERPINI1*[35,36] (Fig. 2g). To validate this finding, we next examined a panel of androgen activated (Supplementary Fig. 2f, g) and androgen repressed genes (Supplementary Fig. 2h, i) by quantitative PCR (qPCR) in 22RV1 and VCaP cells stimulated with R1881, and a similar trend in the expression of these genes was noted. Since, SPINK1 is a secretory protein, we next performed enzyme-linked immunosorbent assay (ELISA) to detect its level in the conditioned media (CM) of 22RV1 cells upon androgen stimulation, and a significant decrease

in the SPINK1 levels both in the CM and total cell lysate (CL) was observed (Supplementary Fig. 2j).

Non-steroidal pharmacological inhibitors for AR, namely bicalutamide (Bic) and enzalutamide (Enza) have been widely used for the treatment of locally advanced non-metastatic and metastatic PCa[21]. Therefore, we determined the effect of these anti-androgens on SPINK1 expression in VCaP cells, and treatment with Enza remarkably increased the *SPINK1* transcript (~4-fold) and protein levels, accompanied with reduced expression of androgen driven-genes namely *KLK3* and *ERG* (Fig. 2h–j). Similarly, a significant increase in the SPINK1 levels both in CM and CL of the Enza-treated VCaP cells was observed by ELISA (Supplementary Fig. 2k). To corroborate these findings, we treated VCaP cells with Bic (25 and 50 μM) and found a significant increase in the SPINK1 expression (Supplementary Fig. 2l–n). Also, a significant increase in the migratory properties of androgen stimulated VCaP cells treated with Bic or Enza was observed (Supplementary Fig. 3a). Since 22RV1 are less responsive to androgen stimulation as compared to VCaP cells, thus we primed the 22RV1 cells either with R1881 (10 nM) or Enza (10 μM) for 3 days, followed by Enza treatment or R1881 stimulation for next 3 days (Fig. 2k). As anticipated, blocking androgen signaling with Enza in the androgen-primed 22RV1 cells result in significant increase in SPINK1 expression, while Enza-treated 22RV1 cells stimulated with R1881 show a repression of SPINK1 (Fig. 2l). To examine the effect of long-term DHT treatment on *SPINK1* expression, 22RV1 cells were cultured in DHT (8 nM) for 2 months, which resulted in more

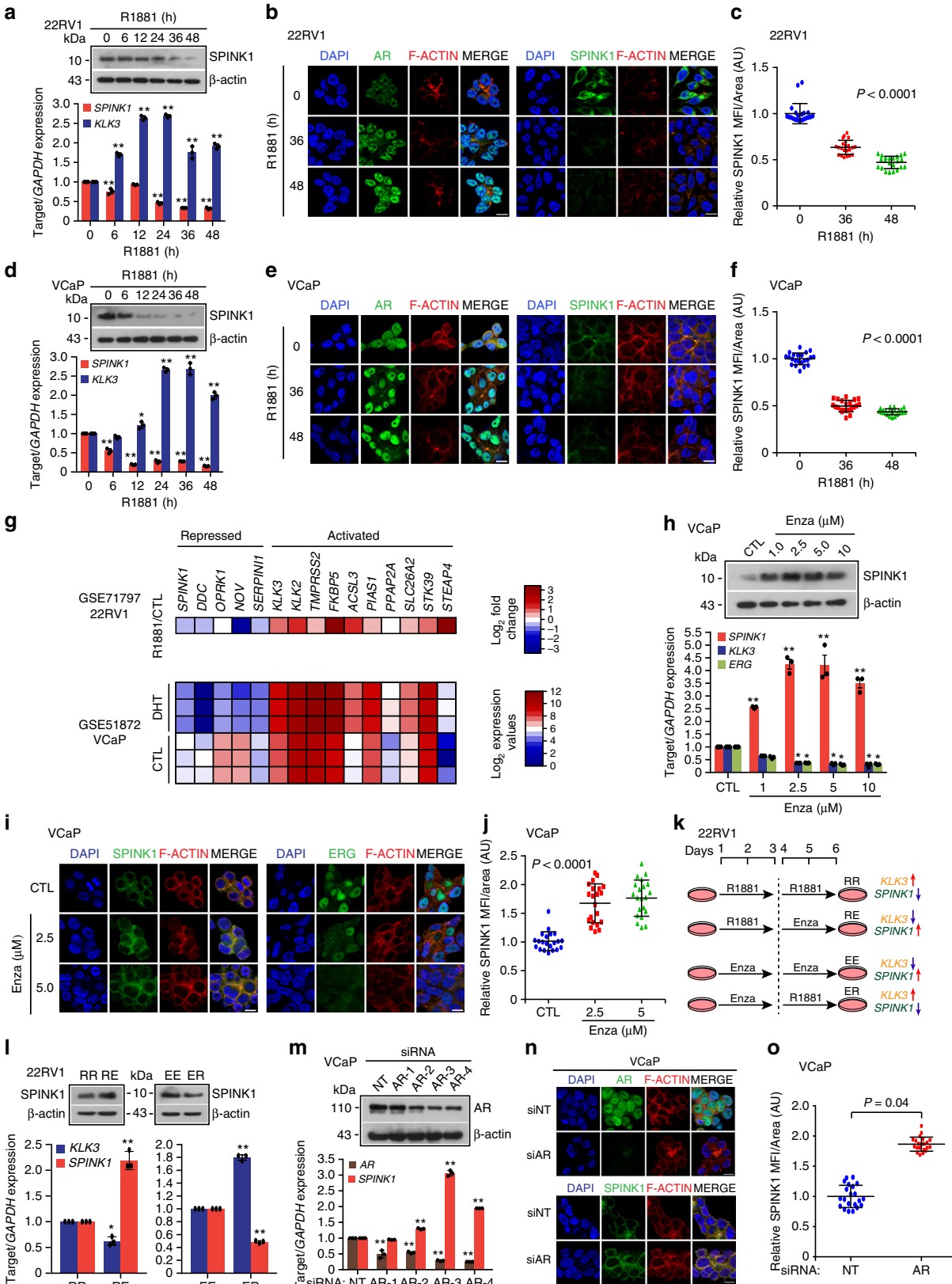

than ~80% reduction in *SPINK1* expression (Supplementary Fig. 3b). Conversely, long-term blockade of androgen signaling in 22RV1 cells using Bic (5 μM) led to significant increase (~1.5 folds) in the *SPINK1* expression (Supplementary Fig. 3c). Similar results were obtained in CWR22Pc cells, a derivative cell line of CWR22 xenograft subjected to long-term Bic treatment (Supplementary Fig. 3d).

As an alternative to pharmacological inhibition of AR signaling, we used small interfering RNA (siRNA) to abolish AR expression in 22RV1 and VCaP cells and examined any

**Fig. 2 Androgen signaling negatively regulates SPINK1 expression in prostate cancer. a** Immunoblot for SPINK1 in 22RV1 cells stimulated with R1881 (10 nM) (top). QPCR data showing relative expression of *SPINK1* and *KLK3* in the same cells (bottom). **b** Immunostaining for SPINK1 and AR in 22RV1 cells stimulated with R1881 (10 nM). **c** Same as **b**, except dot plot represents quantification for SPINK1 mean fluorescence intensity (MFI) per unit area shown as arbitrary units (AU). **d** Same as **a**, except VCaP cells were used. **e** Same as **b**, except VCaP cells were used. **f** Same as **c**, except quantification of VCaP cells as depicted in **e**. **g** Heatmap depicting relative expression of androgen regulated genes in androgen stimulated 22RV1 (top, GSE71797) and VCaP cells (bottom, GSE51872). **h** Immunoblot showing SPINK1 expression in VCaP cells treated with enzalutamide (top). QPCR data showing relative expression of *SPINK1*, *KLK3*, and *ERG* (bottom). **i** Immunostaining for SPINK1 and ERG using same cells as **h**. **j** Same as **i**, except dot plot represents quantification for SPINK1 fluorescence intensity. **k** Schema depicting sequential treatment of 22RV1 cells with R1881 (10 nM) and enzalutamide (10 μM). **l** Immunoblot showing SPINK1 expression in 22RV1 cells as indicated in **k** (top). QPCR data showing relative expression of *SPINK1* and *KLK3* using same cells in **k** (bottom). **m** Immunoblot for AR in *AR*-silenced and control VCaP cells (top). QPCR data showing relative expression of *AR* and *SPINK1* using same cells (bottom). **n** Immunostaining for AR and SPINK1 using same cells as **m**. **o** Same as **n**, except dot plot represents quantification for SPINK1 fluorescence intensity. For panels **b**, **e**, **i**, **n**, scale bar represents 10 μm. For panels **c**, **f**, **j**, **o**, data represents mean ± SD using ten fields per experimental condition. For panels **a**, **d**, **h**, **l**, **m**, experiments were performed with *n* = 3 biologically independent samples; data represents mean ± SEM. For panels **a**, **d**, **h**, **m** two-way ANOVA, Dunnett's multiple-comparisons test; (**c**, **f**, **j**) one-way ANOVA, Tukey's multiple-comparisons test; (**l**) two-way ANOVA, Sidak's multiple-comparisons test; (**o**) two-tailed unpaired Student's *t*-test was applied. *$P \leq 0.05$ and **$P \leq 0.001$. Source data for **a**, **d**, **h**, **l**, **m** are provided as a Source Data file.

change in SPINK1 levels. Similar to the small molecule inhibition of AR signaling, siRNA-mediated *AR*-silenced 22RV1 cells also exhibit moderate increase in the expression of SPINK1 (Supplementary Fig. 3e–h), while a robust increase (~3-fold) in the SPINK1 was observed in *AR*-silenced VCaP cells (Fig. 2m–o and Supplementary Fig. 3i). Furthermore, siRNA mediated knockdown of AR splice-variants (*AR-V1*, *AR-V3*, *AR-V4*, and *AR-V7*) in 22RV1 cells (GSE80743) led to an increase in *SPINK1* expression compared to control (Supplementary Fig. 3j). Taken together, our findings demonstrate that AR signaling negatively regulates SPINK1 expression and draws attention to AR antagonists mediated upregulation of SPINK1 in prostate cancer.

**AR directs transcriptional repression of *SPINK1* in PCa.** The role of AR has been extensively characterized both as a transcriptional activator, as well as a repressor[35]. To examine whether AR directly regulates *SPINK1* transcription we analyzed the presence of putative AR binding sites in the *SPINK1* promoter region, and scanned the region for the presence of androgen response elements (AREs) by employing publicly available transcription factor binding prediction software, JASPAR (http://www.jaspar.genereg.net) and MatInspector (http://www.genomatix.de). Several putative AREs within the ~5 kb region upstream of transcription start site (TSS) of *SPINK1* were identified (Fig. 3a). Further, analysis of the publicly available Chromatin Immunoprecipitation-Sequencing (ChIP-Seq) dataset for AR binding in androgen stimulated VCaP cells (GSE58428) revealed another putative ARE on the *SPINK1* promoter (Fig. 3b).

To confirm AR binding on the *SPINK1* promoter, we performed ChIP-quantitative PCR (ChIP-qPCR) for AR in R1881-stimulated 22RV1 cells, and a significant enrichment for AR-binding at three distinct sites (*ARE-1*, *ARE-2*, and *ARE-3*) was observed (Fig. 3c and Supplementary Fig. 4a). Promoters for *KLK3* and *NOV* were used as positive controls[36]. To determine the transcriptional activity of *SPINK1* upon androgen stimulation, we performed ChIP-qPCR for the C-terminal domain (CTD) of the largest subunit of RNA polymerase II (Pol-II), transcription initiation specific Pol-II CTD Ser5 phosphorylation (p-Pol-II-S5) and transcription elongation specific Pol-II CTD Ser2 phosphorylation (p-Pol-II-S2)[37]. Interestingly, a significant decrease in the recruitment of total Pol-II accompanied with a remarkable reduction in the occupancy of p-Pol-II-S5 and p-Pol-II-S2 on the *SPINK1* promoter was observed, indicating its transcriptional repression in androgen stimulated 22RV1 cells (Fig. 3d). A similar trend was noted on the *NOV* promoter, while an increased occupancy of p-Pol-II-S2 and no significant change in the total Pol-II and p-Pol-II-S5 recruitment on the *KLK3*

promoter was observed (Supplementary Fig. 4b-d). Moreover, a significant reduction in the enrichment of H3K9Ac activation marks on the *SPINK1* promoter was observed in R1881-stimulated 22RV1 cells, which further confirms its transcriptionally repressed-state (Fig. 3e). Conversely, enrichment of H3K9Ac marks on *KLK3* promoter indicates its transcriptionally active state (Fig. 3e). No change in the levels of total Histone H3 at the *SPINK1* and *KLK3* promoters was observed (Supplementary Fig. 4e). We next examined for any change in AR recruitment on the *SPINK1* promoter in Enza-treated VCaP cells, and observed a remarkable decrease in the AR recruitment, indicating impaired AR-binding (Fig. 3f). No change in Pol-II occupancy on the *SPINK1* promoter was observed in R1881-stimulated VCaP cells (Supplementary Fig. 4f). Next, to investigate whether *SPINK1* promoter is in transcriptionally poised-state, we examined for the presence of the H3K27me3 (repressive) and H3K4me3 (activation) histone marks in VCaP cells[38]. A significant gain in H3K27me3, while no change in H3K4me3 marks were found, confirming the poised state of *SPINK1* promoter (Supplementary Fig. 4g, h). A similar pattern in the repressive/activation marks was also observed for *NOV*; conversely, *KLK3* being transcriptionally active, exhibit enrichment of H3K4me3 and no change in H3K27me3 marks (Supplementary Fig. 4g, h).

To further confirm the AR signaling-mediated transcriptional repression of *SPINK1*, we performed luciferase reporter assay using proximal (SPINK1-PP) and distal (SPINK1-DP) promoter regions of *SPINK1* in 22RV1 cells. A concentration dependent decrease in the luciferase activity was observed in 22RV1 cells transfected with SPINK1-PP and SPINK1-DP upon androgen stimulation (Fig. 3g). A significant increase in the luciferase activity of both the reporter constructs was observed upon Enza treatment (Fig. 3h). The PSA (*KLK3*) promoter construct was used as a positive control for androgen stimulation (Supplementary Fig. 4i, j). Similarly, siRNA mediated knockdown of *AR* also led to a significant increase in the reporter activity of the SPINK1-PP (Supplementary Fig. 4k). Further, we mutated ARE (ARE MT) in the SPINK1-DP construct and performed luciferase assay, and as a result, no change in the luciferase activity was recorded in the 22RV1 cells transfected with mutant SPINK1-DP (Fig. 3i). Furthermore, 22RV1 cells transfected with wildtype or mutant AR (ΔNLS and V581F) show significant decrease in the luciferase activity of both SPINK1-PP and SPINK1-DP with wildtype AR, while no change was observed with AR mutants (Fig. 3i). Together these findings indicated that the AR acts as a direct transcriptional repressor of *SPINK1*, and attenuating AR signaling using AR-antagonists relieve *SPINK1* transcriptional repression resulting in its upregulation (Fig. 3j).

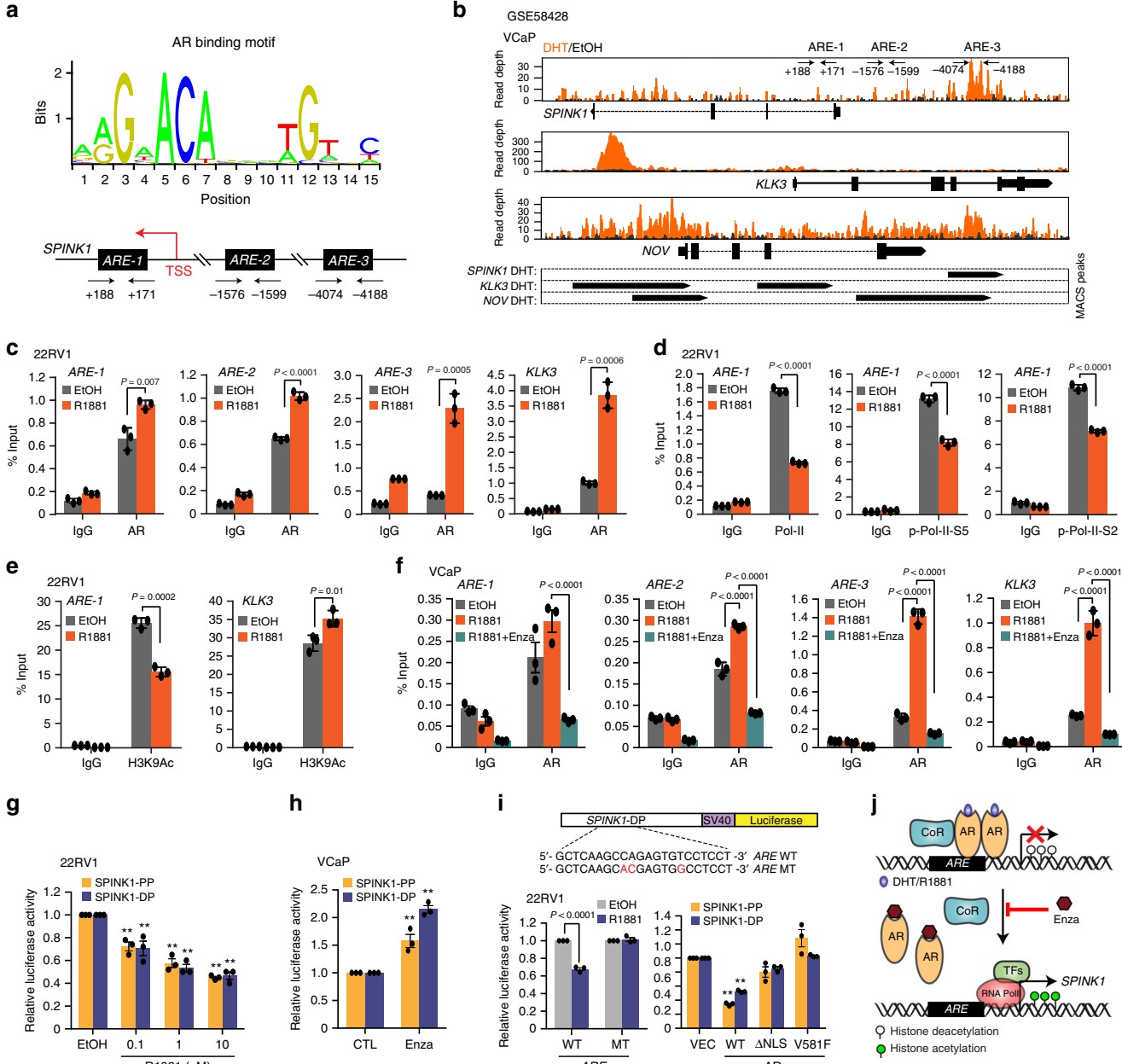

**Fig. 3 AR directly binds to *SPINK1* promoter region and modulates its expression. a** Schema showing AR binding motif obtained from JASPAR database (top). Bottom panel showing genomic location for the AREs on the *SPINK1* promoter. **b** ChIP-Seq profiles indicating AR enrichment on the *SPINK1*, *KLK3*, and *NOV* gene loci in androgen stimulated VCaP cells (GSE58428). Bottom panel indicates the MACS identified peaks for AR binding on the promoters of *SPINK1*, *KLK3,* and *NOV*. **c** ChIP-qPCR data showing recruitment of AR on the *SPINK1* promoter upon R1881 (10 nM) stimulation in 22RV1 cells. *KLK3* promoter was used as a positive control for the androgen stimulation experiment. **d** Same as in **c**, except total RNA Pol-II, p-Pol-II-Ser5 and p-Pol-II-Ser2 on the *SPINK1* promoter. **e** Same as in **c**, except H3K9Ac (H3 lysine 9 acetylation) marks on the *SPINK1* and *KLK3* promoters. **f** ChIP-qPCR data depicting enrichment of AR on the *SPINK1* and *KLK3* promoters in R1881 (10 nM) stimulated VCaP cells treated with or without enzalutamide (10 μM). **g** Luciferase reporter activity of the proximal (SPINK1-PP) and distal SPINK1 (SPINK1-DP) promoters in R1881(10 nM) stimulated 22RV1 cells. **h** Same as in **g** except enzalutamide (10 μM) treated VCaP cells were used. **i** Schematic showing luciferase reporter constructs with SPINK1-DP wild-type (WT) or mutated (MT) ARE sites (altered residues in red) (top). Bar plots showing luciferase reporter activity of SPINK1-DP WT or MT in R1881 stimulated (10 nM) 22RV1 cells (bottom, left) and 22RV1 cells co-transfected with SPINK1-PP or SPINK1-DP and control vector (VEC), AR wildtype (WT) or AR mutants (ΔNLS and V581F) constructs (bottom, right). **j** Illustration showing AR signaling mediated regulation of *SPINK1* in prostate cancer, wherein CoR (corepressor), TFs (transcription factors) and Enza (enzalutamide) is shown. Experiments were performed with $n = 3$ biologically independent samples; data represents mean ± SEM. For panels **c**, **d**, **e** two-tailed unpaired Student's $t$-test; **g**, **i** two-way ANOVA, Dunnett's multiple-comparisons test; **f**, **h** two-way ANOVA, Sidak's multiple-comparisons test was applied. *$P \leq 0.05$ and **$P \leq 0.001$.

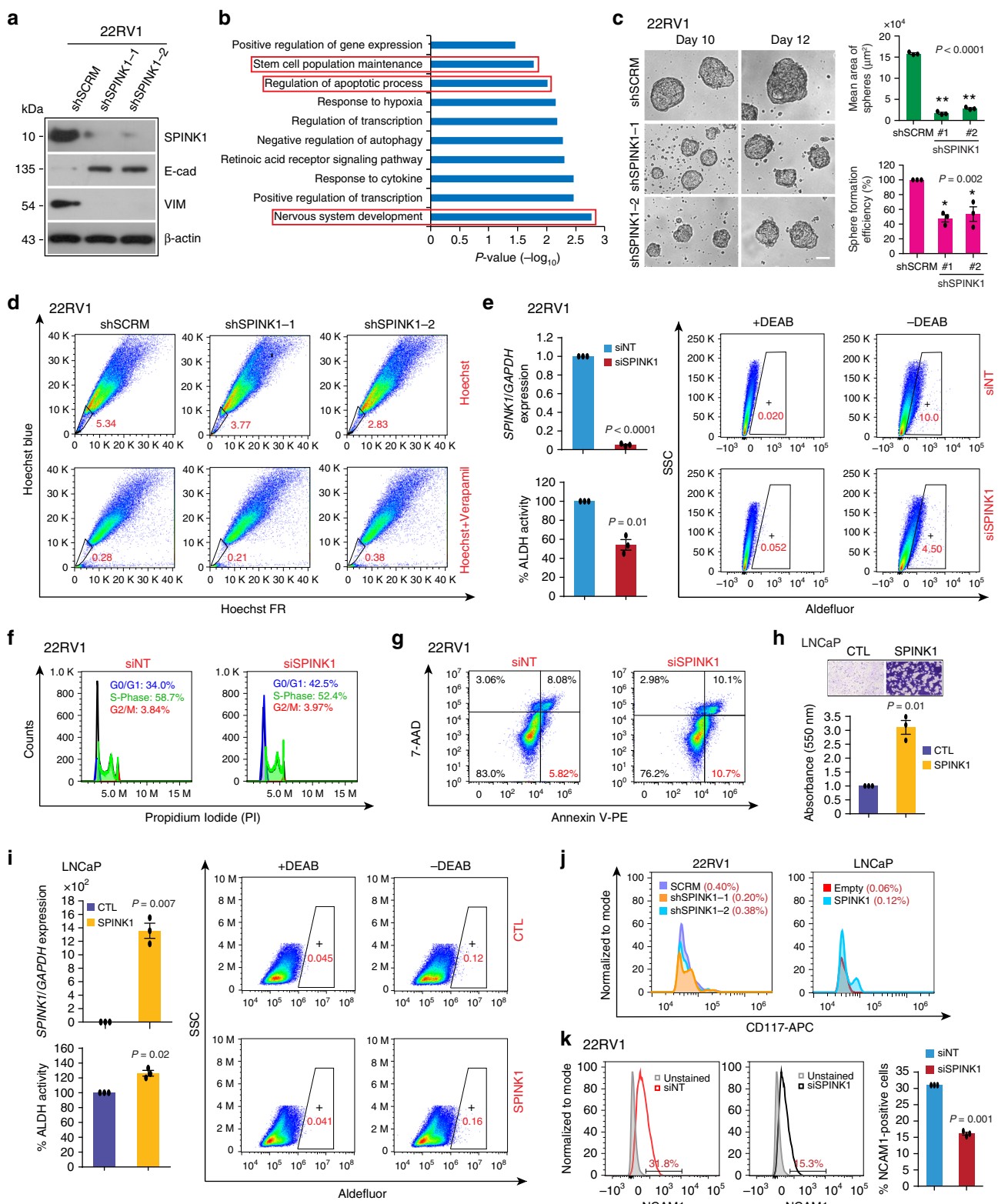

**SPINK1 promotes cellular plasticity and stemness in PCa.**
Blockade of androgen signaling in PCa is known to induce EMT and stemness[16]. Moreover, SPINK1, as a pro-proliferative and pro-invasive factor has been implicated in metastasis and chemoresistance[11,12]. To identify the role of SPINK1 in EMT, we established stable *SPINK1*-silenced 22RV1 cells (22RV1-shSPINK1) and scrambled control (22RV1-shSCRM) using

lentivirus-based short-hairpin RNAs (Fig. 4a and Supplementary Fig. 5a, b), and examined well-known EMT markers. Intriguingly, a significant increase in the E-cadherin (epithelial marker) and a decrease in Vimentin (mesenchymal marker) expression was observed in 22RV1-shSPINK1 cells as compared to 22RV1-shSCRM, highlighting the role of SPINK1 in EMT (Fig. 4a and Supplementary Fig. 5c). Previously, SPINK1 has been implicated

**Fig. 4 SPINK1 promotes EMT, stemness and chemoresistance in prostate cancer. a** Immunoblot analysis for SPINK1, E-Cadherin and Vimentin levels in stable *SPINK1*-silenced (shSPINK1-1 and shSPINK1-2) and control (shSCRM) 22RV1 cells. **b** Biological pathways downregulated in 22RV1-shSPINK1 cells relative to 22RV1-shSCRM cells obtained by DAVID analysis. Bars represent $-\log_{10}$ (*P*-values). **c** Representative phase contrast microscopic images for the prostatospheres using same cells as **a** (left). Bar plots depict mean area and percent sphere formation efficiency of the prostatospheres (right). Scale bar represents 100 μm. **d** Hoechst-33342 staining for the side population analysis using same cells as **a**. Data was analyzed by putting blue and far-red filters, gated regions are marked in red for each panel. **e** QPCR data showing relative expression of *SPINK1* in siRNA-mediated *SPINK1*-silenced 22RV1 cells (top, left). Quantification of ALDH activity using flow cytometry using same cells. Flow cytometric graphs showing fluorescence intensity of catalyzed ALDH substrate in the presence or absence of DEAB. Marked windows indicate percentage of ALDH1 positive cell population. **f** Flow cytometric analysis demonstrating cell-cycle arrest in *SPINK1*-silenced 22RV1 cells using propidium iodide (PI) DNA staining. **g** Flow cytometry analysis depicting cell apoptosis by Annexin V-PE and 7-AAD staining using same cells as **e**. **h** Transwell migration assay using stable SPINK1 overexpressing and control LNCaP cells. Top panel shows the representative microphotographs (scale bar = 100 μm) and bar plot depicts the data quantification (bottom). **i** Same assay as **e**, except LNCaP cells with transient overexpression of SPINK1 was used. **j** Flow cytometry analysis showing CD117-APC (c-KIT) expression using same cells as **a** and **h**. **k** Flow cytometry histograms depicting the expression of CD56-PE/Cy7 (NCAM1) using same cells as **e**. The cell populations were normalized to mode. Bar plot represents relative surface expression of NCAM1. Experiments were performed with $n = 3$ biologically independent samples; data represents mean ± SEM. For panel **c** one-way ANOVA, Dunnett's multiple-comparisons test; **e**, **h**, **i**, **k** two-tailed unpaired Student's *t*-test was applied. *$P \leq 0.05$ and **$P \leq 0.001$. The source data for **a** is provided as a Source Data file.

in chemoresistance in colorectal cancer[12], we thus investigated whether SPINK1 governs similar attribute in PCa. As expected, a significant increase in sensitivity towards established chemotherapeutic drugs, namely doxorubicin, cisplatin and 5-fluorouracil was recorded in 22RV1-shSPINK1 cells as compared to control (Supplementary Fig. 5d–f).

To identify the biological processes governed by SPINK1 and elucidate its functional relevance, we performed microarray-based gene expression profiling of 22RV1-shSPINK1 and 22RV1-shSCRM cells. Our analysis revealed 697 genes downregulated in 22RV1-shSPINK1 cells ($\log_2$ fold change > 0.5 or < −0.5, 90% confidence interval), which were further analyzed for enriched pathways ($P < 0.05$) using DAVID (Database for Annotation, Visualization and Integrated Discovery). Notably, genes downregulated upon *SPINK1* knockdown were associated with critical pathways, namely, stem-cell maintenance, apoptosis and nervous system development (Fig. 4b and Supplementary Table 1). Next, to examine the self-renewal ability of 22RV1-shSPINK1 cells, we performed prostatosphere assay, and observed a significant decrease in the number and size of the prostatospheres (Fig. 4c). Furthermore, we executed the side-population (SP) assay to evaluate the efflux of Hoechst dye via ABC-transporters[39], a significant reduction ~29% and ~47% in the SP was noted in 22RV1-shSPINK1-1 and 2 cells, respectively (Fig. 4d). Since, aldehyde dehydrogenase (ALDH) is crucial for promoting stemness and chemoresistance in cancer[40], we performed ALDH assay, and found a significant decrease in its activity in the siRNA-mediated *SPINK1*-silenced 22RV1 cells (Fig. 4e). Moreover, cell cycle arrest in G0/G1 phase and apoptosis was also observed in these cells (Fig. 4f, g). Conversely, ectopic overexpression of SPINK1 in LNCaP cells show a robust increase in the migratory properties (Fig. 4h and Supplementary Fig. 5g) In addition, an increase in the ALDH activity was observed in SPINK1 overexpressing LNCaP cells (Fig. 4i). We also examined CD117 (c-KIT), a tyrosine kinase receptor associated with cancer progression and stem-cell maintenance[16], and a reduction in the percent c-KIT positive cells was observed in the 22RV1-shSPINK1 cells, while an increase was noted in SPINK1 overexpressing LNCaP cells (Fig. 4j). Taken together, these findings highlight the predominant role of SPINK1 in EMT, stemness and drug resistance in prostate cancer.

As shown in Fig. 4b, DAVID analysis revealed nervous system development as one of the most enriched GO terms, hence, we next investigated the expression of NEPC markers (*SYP*, *CHGA*, and *ENO2*) in 22RV1-shSPINK1 cells relative to control. Of these, a significant decrease was observed only in the expression of *SYP* (Supplementary Fig. 5h). Nevertheless, siRNA-mediated

*SPINK1*-silenced 22RV1 cells show a significant reduction in the surface expression of the neural cell adhesion molecule-1 (NCAM1), an established marker of neural lineage and neurite outgrowth (Fig. 4k). These findings accentuate the plausible role of SPINK1 in driving cellular plasticity and its association with neuroendocrine (NE) phenotype.

**SPINK1 upregulation is associated with NE phenotype in PCa**. To understand the effect of long-term androgen deprivation on *SPINK1* expression, we analyzed publicly available gene expression dataset (GSE8702), wherein LNCaP cells (SPINK1-negative) were androgen deprived for 12 months. Remarkably, with prolonged androgen deprivation, a robust increase in the *SPINK1* expression was noticed (Fig. 5a). Furthermore, Gene Set Enrichment Analysis (GSEA) of these cells revealed a significant decrease in the expression of genes associated with androgen-signaling and positive enrichment of the pathways associated with neuron markers and axon guidance (Supplementary Fig. 6a), thus, emphasizing the probable role of SPINK1 in cellular plasticity and NE-like morphology. To confirm the association of SPINK1 upregulation with NE-transdifferentiation, LNCaP cells were cultured in androgen-deprived condition for 30 days (Fig. 5b). Consistent with previous report[41], we observed a gradual change in the morphology of LNCaP cells, from an epithelial to a more NE-like phenotype (LNCaP-AI), an androgen-independent cell line, exhibiting neuron-like projections with a concomitant increase in the NEPC markers namely, SYP, CHGA, ENO2, and NCAM1, and a significant decrease in PSA and REST (Fig. 5b, d and Supplementary Fig. 6b). Intriguingly, in LNCaP-AI and in long-term androgen-deprived C4-2 cells, a LNCaP derivative, show a remarkable increase in SPINK1 both at transcript and protein levels with concomitant drop in *KLK3* (Fig. 5c–e and Supplementary Fig. 6c). Moreover, LNCaP-AI cells also show a significant increase in the expression of EMT (*NCAD*, *VIM*, and *TWIST1*) and stemness markers (*SOX2*, *CD44*, and *KIT*) (Fig. 5d and Supplementary Fig. 6d, e). Next, we examined for any possible association between *SPINK1* expression and NEPC markers in TCGA-PRAD and MSKCC PCa patient cohorts. Interestingly, a positive correlation between *SPINK1* expression and NEPC markers (*SYP*, *MYCN*, and *CHGB*) was observed (Supplementary Fig. 6f, g), supporting the plausible role of SPINK1 in cellular plasticity and reprogramming.

Since, we observed a remarkable increase in SPINK1 expression with reduced AR signaling in LNCaP-AI cells, we next examined the effect of constitutively active AR signaling by generating doxycycline-inducible AR-V7 (AR-splice-variant 7)

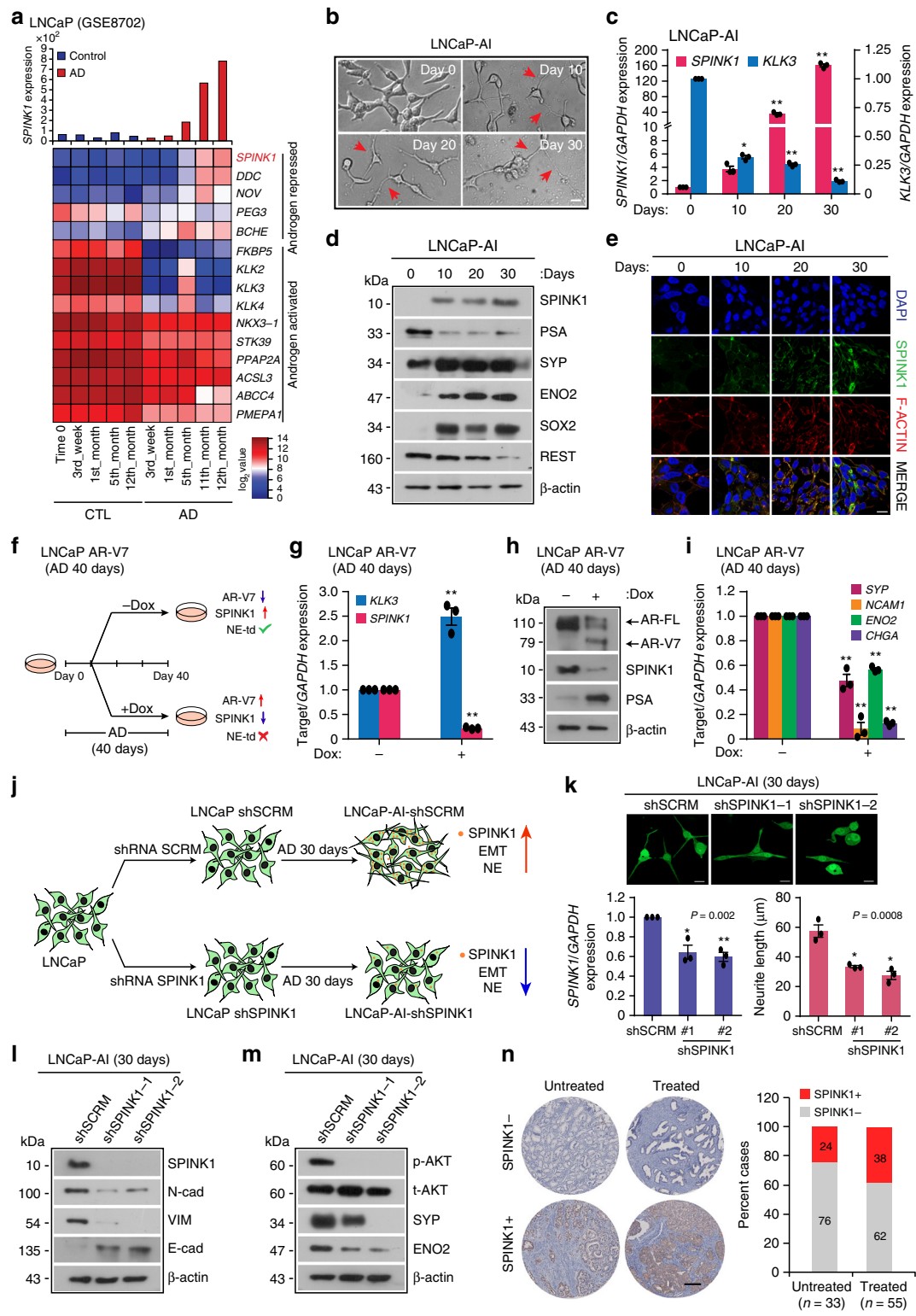

LNCaP cells, and subjecting them to long-term androgen deprivation (40 days) with or without induction at day 10 (Fig. 5f). A robust increase in PSA level confirms re-activation of AR signaling via AR-V7 overexpression in androgen-deprived doxycycline-induced LNCaP AR-V7 cells as compared to uninduced cells (Fig. 5g, h). Intriguingly, a concomitant decrease in the SPINK1 expression was observed in induced LNCaP AR-V7 cells compared to control, reaffirming the AR-mediated

regulation of SPINK1 (Fig. 5g, h). Furthermore, doxycycline-induced LNCaP AR-V7 cells show reduced expression of NEPC (*SYP, CHGA, ENO2,* and *NCAM1*) (Fig. 5i), EMT (*VIM* and *TWIST1*) and stemness (*SOX2, CD44,* and *KIT*) markers with respect to uninduced control (Supplementary Fig. 6h, i), indicating that reactivation of AR signaling negatively regulates *SPINK1* and hampers NE-transdifferentiation in androgen-deprived LNCaP cells.

**Fig. 5 Androgen-deprivation upregulates SPINK1 in NE-transdifferentiated PCa cells. a** Bar graph showing *SPINK1* expression (top) and heatmap of AR-signaling associated genes including *SPINK1* in long-term androgen deprived (AD) LNCaP cells (GSE8702). **b** Representative phase-contrast images of androgen-deprived LNCaP cells (LNCaP-AI). Red arrow-heads indicate neurite outgrowth. **c** QPCR data showing relative expression of *SPINK1* and *KLK3* using same cells as **b**. **d** Immunoblot assay for SPINK1, PSA, SYP, ENO2, SOX2 and REST using same cells as in **b**. **e** Immunostaining for SPINK1 using same cells as in **b**. **f** Schematic representation of NE-transdifferentiation (NE-td) using doxycycline (dox)-inducible AR-V7 overexpressing LNCaP cells subjected to androgen deprivation (AD) with or without induction (40 ng/ml) at day 10 and cultured upto 30 days. **g** QPCR data showing relative expression of *SPINK1* and *KLK3* using same cells as **f**. **h** Immunoblot assay for AR, AR-V7, SPINK1, and PSA using same cells as **f**. **i** QPCR data showing relative expression of *SYP, NCAM1, ENO2* and *CHGA* using the same cells as **f**. **j** Schema describing generation of LNCaP-AI-shSPINK1 and LNCaP-AI-shSCRM cells by subjecting stable LNCaP-shSPINK1 and LNCaP-shSCRM cells to androgen deprivation (AD) for 30 days. **k** Representative images for the neurite outgrowths in LNCaP-AI-shSCRM cells and LNCaP-AI-shSPINK1 as **j** (top). QPCR data showing relative expression of *SPINK1* (bottom, left) and measurement of neurite outgrowth (bottom, right). **l** Immunoblot analysis for SPINK1, E-Cad, VIM, and N-Cad expression using same cells as **j**. **m** Same as in **l**, except phospho (**p**) and total (**t**) AKT, SYP and ENO2 expression. **n** Representative IHC images for SPINK1 in SPINK1-negative (SPINK1−, top) and SPINK1-positive (SPINK1+, bottom) PCa tumor cores of the VPC tissue microarray (scale bar = 200 µm). Bar plot showing percentage cases of SPINK1 in untreated (*n* = 33) and neoadjuvant-hormone therapy (NHT) treated patients (*n* = 55). Experiments were performed with *n* = 3 biologically independent samples; data represents mean ± SEM. For panels **b**, **e**, **k** scale bar represents 20 µm. For panel **c** two-way ANOVA, Dunnett's multiple-comparisons; **g**, **i** two-way ANOVA, Sidak's multiple-comparisons test; **k** one-way ANOVA, Dunnett's multiple-comparisons test were applied. *$P \leq 0.05$ and **$P \leq 0.001$. Source data for **d**, **h**, **l**, **m** are provided as a Source Data file.

To further confirm the role of SPINK1 in NE-transdifferentiation of LNCaP cells, we established stable *SPINK1*-silenced LNCaP cells (LNCaP-shSPINK1) and scrambled control (LNCaP-shSCRM) using lentivirus-based short-hairpin RNAs, and cultured them in androgen-deprived condition for 30 days (Fig. 5j). Phenotypically androgen-deprived LNCaP-AI-shSPINK1 show reduction in the length of neurite-like projections, indicating altered NE-transdifferentiation (Fig. 5k). Intriguingly, LNCaP-AI-shSPINK1 cells exhibit decrease in the markers for EMT (E-Cad, Vimentin and N-Cad) and NEPC (SYP and ENO2) as compared to LNCaP-AI-shSCRM cells (Fig. 5l, m), indicating the significance of SPINK1 in NE-transdifferentiation and cellular plasticity. Previous studies indicate the role of AKT signaling in advancement of PCa to poorly differentiated small cell prostate carcinoma[42]. Moreover, several studies have shown a critical role of SPINK1 in activating PI3K-AKT signaling cascade in multiple SPINK1-positive cancers[11,12,15]. In concordance to these reports, we also observed a remarkable decrease in AKT signaling in LNCaP-AI-shSPINK1 cells as compared to control cells (Fig. 5m).

To further confirm the significance of SPINK1 in governing the cellular plasticity, we used LNCaP-derived CRPC cell line, namely 16D$^{CRPC}$, and its derivative 42D$^{ENZR}$ and 42F$^{ENZR}$ cell lines established via multiple serial transplantation of the enzalutamide-resistant tumors in athymic male mice[29]. Enzalutamide-resistant cell lines harbor reduced AR activity as depicted by the minimal expression level of PSA as compared to parental 16D$^{CRPC}$ cells (Supplementary Fig. 7a). Furthermore, GSEA plots using the RNA-seq data of 16D$^{CRPC}$ and 42D$^{ENZR}$ cells reveal reduced expression of genes associated with AR signaling, with concomitant increase in the expression of neuronal markers and genes-associated with neurogenesis (Supplementary Fig. 7b). Moreover, 42D$^{ENZR}$ and 42F$^{ENZR}$ cells show higher expression of NEPC markers namely *SYP, CHGA,* and *ENO2* along with significant increase in SPINK1 levels as compared to 16D$^{CRPC}$ cells (Supplementary Fig. 7c–e). Transcriptomic analysis of these cells revealed negative association of *SPINK1* expression with AR signaling associated genes[29,43] (Supplementary Fig. 7f). Interestingly, siRNA-mediated knockdown of *SPINK1* in 42D$^{ENZR}$ and 42F$^{ENZR}$ cells results in a significant decrease in the expression of *SYP*, while reduced *CHGA* level in only 42F$^{ENZR}$ cells was noted (Supplementary Fig. 7g, h). Taken together, these findings highlight the critical role of SPINK1 in the maintenance of NE-phenotype.

To investigate the effect of ADT on SPINK1 in PCa patients administered with neoadjuvant hormone therapy (NHT), we examined the expression of SPINK1 in a TMA comprising of PCa specimens (*n* = 88) by performing IHC staining, wherein 55 out of 88 patients were given NHT for 3 months. In line with our in vitro findings, ~38% (21 out of 55) patients who underwent NHT exhibit SPINK1 positive status compared to only ~24% (8 out of 33) in the untreated group (Fig. 5n). Although, ADT or NHT-mediated SPINK1 upregulation and associated risk-factors need to be tested in a larger PCa patients' cohort. Collectively, our findings suggest that androgen-deprivation therapies may have an adverse effect, and the benefits must be weighed against treatment. Conclusively, we also show that elevated SPINK1 levels during NE-transdifferentiation strongly emphasizes the potential role of SPINK1 in governing stemness and cellular plasticity in prostate cancer.

**Expression of *SPINK1* is modulated by SOX2 and REST in PCa.** The role of SRY (sex determining region Y)-box 2 (SOX2) has been implicated in NE-differentiation and reprogramming/lineage plasticity in *RB1* and *TP53* deficient PCa[44]. Since, SOX2 is a known androgen repressed gene[45], and our data also show reduced *SOX2* expression in androgen-stimulated 22RV1 cells (Supplementary Fig. 8a), we sought to examine SOX2-mediated regulation of *SPINK1*. We scanned the *SPINK1* promoter for SOX2 binding motif using MatInspector, and identified three putative binding sites (*S1, S2,* and *S3*) (Fig. 6a and Supplementary Fig. 8b). To investigate that SPINK1 upregulation in LNCaP-AI cells is mediated through SOX2 during NE-transdifferentiation, we examined SOX2 occupancy on the *SPINK1* promoter using these cells and observed a remarkable enrichment of SOX2 at three distinct binding sites (Fig. 6a). Similarly, 22RV1, an endogenously SOX2 positive cell line, also exhibit a significant SOX2 enrichment on the *SPINK1* promoter (Fig. 6b). In addition, an increase in the occupancy of Pol-II was noticed on the *SPINK1* promoter in LNCaP-AI and 22RV1 cells (Fig. 6c, d), signifying its increased transcriptional activity. Furthermore, silencing *SOX2* in these cell lines result in a remarkable reduction in the SPINK1 levels (Fig. 6e, f and Supplementary Fig. 8c). Contrariwise, ectopic SOX2 overexpression in LNCaP cells show a robust increase in the SPINK1 expression (Fig. 6g). Finally, luciferase reporter assay also indicates a significant increase in the luciferase activity of SPINK1-DP promoter in the SOX2 overexpressing LNCaP cells (Fig. 6h), thus reaffirming the SOX2-mediated positive transcriptional regulation of *SPINK1*.

Downregulation of REST, a transcriptional co-repressor of AR, plays a critical role in the progression of CRPC to NEPC[31,46].

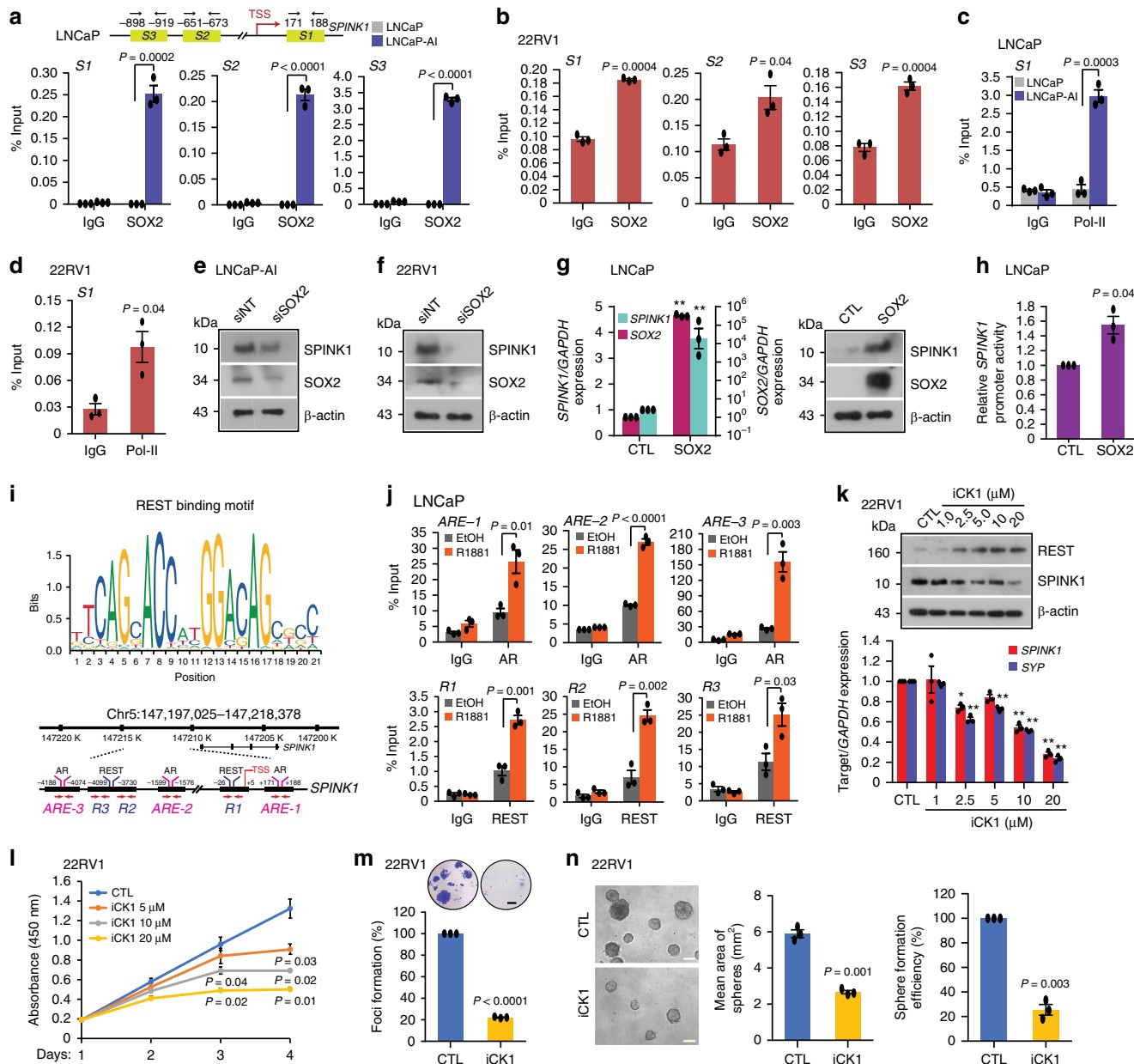

**Fig. 6 Reprogramming factor SOX2 and AR transcriptional co-repressor REST modulate *SPINK1* expression. a** Schematic showing SOX2 binding elements (*S1*, *S2*, and *S3*) on the *SPINK1* promoter (top). ChIP-qPCR data for SOX2 occupancy on the *SPINK1* promoter in wildtype LNCaP and LNCaP-AI cells (androgen-deprived for 15 days) (bottom). **b** Same as in **a**, except 22RV1 cells. **c** ChIP-qPCR data for RNA Pol-II binding on the *SPINK1* promoter using cells as **a**. **d** Same as **c**, except 22RV1 cells. **e** Immunoblot for SOX2 and SPINK1 in siRNA mediated *SOX2*-silenced LNCaP-AI and control cells. **f** Same as **e**, except 22RV1 cells. **g** QPCR data showing relative expression of *SOX2* and *SPINK1* upon transient SOX2 overexpression in LNCaP cells (left). Immunoblot for SOX2 and SPINK1 expression(right). **h** Luciferase reporter activity of the SPINK1 distal-promoter (SPINK1-DP) using same cells as **g**. **i** REST binding motif obtained from JASPAR (top). Genomic location for AR and REST binding on the *SPINK1* promoter (bottom). **j** ChIP-qPCR data showing AR and REST occupancy on the *SPINK1* promoter in R1881 stimulated (10 nM) LNCaP cells. **k** Immunoblot for the REST and SPINK1 levels in 22RV1 cells treated with Casein Kinase 1 inhibitor (iCK1) as indicated (top). QPCR data for relative *SPINK1* and *SYP* expression (bottom). **l** Cell proliferation assay using 22RV1 cells treated with different concentrations of iCK1. **m** Foci formation assay using 22RV1 cells treated with iCK1 (20 μM). Inset showing representative images depicting foci (scale bar: 500 μm). **n** Representative phase contrast microscopic images of 3D tumor spheroids using same cells as **m** (left). Bar plots depict mean area and efficiency of the sphere formation. Scale bar represents 1000 μm. Experiments were performed with n = 3 biologically independent samples; data represents mean ± SEM. For panels **a**–**d**, **h**, **j**, **m**–**n** two-tailed unpaired Student's *t*-test; **g** two-way ANOVA, Sidak's multiple-comparisons test; **k** two-way ANOVA, Dunnett's multiple-comparisons test were applied. *$P \leq 0.05$ and **$P \leq 0.001$. Source data for **e**, **f**, **g**, **k** are provided as a Source Data file.

Having established the role of AR signaling in *SPINK1* regulation and NE-transdifferentiation, we next examined the plausible association of SPINK1 with REST and its other complex members in TCGA-PRAD and MSKCC cohorts. We employed quartile-based normalization method[34] to stratify the patients based on

high and low *SPINK1* expression. Notably SPINK1-high patients (SPINK1-positive) show inverse correlation between *SPINK1* and *REST*, as well as other members of its complex namely *RCOR1*, *SIN3A*, *HDAC1* (Supplementary Fig. 8d, e). In agreement androgen stimulation in 22RV1, LNCaP, and VCaP cells result

in a significant increase in the REST levels (Supplementary Fig. 8f), while treating VCaP cells with AR-antagonists resulted in reduced REST expression (Supplementary Fig. 8g). To investigate whether REST is acting as a transcriptional co-repressor of AR in SPINK1 regulation, we examined SPINK1 promoter for the REST binding motif within ~5 kb region of the TSS using MatInspector (Fig. 6i). A robust enrichment of AR at the AREs (ARE-1, ARE-2, and ARE-3) along with REST recruitment at the three distinct RE-1 site (R1, R2, and R3) adjacent to AREs on the SPINK1 promoter was observed in androgen-stimulated LNCaP cells (Fig. 6j and Supplementary Fig. 8h).

In hippocampal neurons, Casein Kinase 1 (CK1) is known to phosphorylate the non-canonical degron motifs in the C-terminal of REST, enabling its binding to the F-box protein E3 ubiquitin ligase SCF (β-TrCP). This, in turn, results in ubiquitin-mediated proteasomal degradation of REST[47,48]. Therefore, we restored REST levels in 22RV1 cells using CK1 inhibitor (iCK1, D4476), and observed a significant increase in the REST levels, with a concomitant decrease in the expression of SPINK1, SYP and other REST target genes (Fig. 6k and Supplementary Fig. 8i). Likewise, ectopic overexpression of REST in 22RV1 cells result in downregulation of SPINK1, while silencing REST in LNCaP cells show an increase in the SPINK1 levels, as well as other REST targets (Supplementary Fig. 8j–l). We next examined whether restoration of REST levels via iCK1 abrogates SPINK1-mediated oncogenic properties, by treating 22RV1 cells with a range of iCK1 concentrations. Intriguingly, a significant reduction in the cell proliferation and number of foci was observed in iCK1 treated 22RV1 cells (Fig. 6l, m). We also observed a significant reduction in the number and size of spheroids in the iCK1-treated 22RV1 cells using three-dimensional tumor spheroid assay (Fig. 6n). Collectively, we have shown the direct role of SOX2 in the transcriptional regulation of SPINK1 in prostate cancer. We also establish that REST acts as a transcriptional corepressor of AR in modulating the SPINK1 expression, thus a cease in AR signaling during NE-transdifferentiation results in SPINK1 upregulation, and its overexpression positively associates with NE-like phenotype.

**ADT upregulates SPINK1 and NE-markers in mice and PCa patients.** To investigate the impact of androgen ablation and mimic the effects of AR antagonists in CRPC, we used castrate-resistant tumors generated by orthotopic implantation of VCaP cells in immunodeficient (HSD/athymic nude–Foxn1[nu]) mice, administered with vehicle (Veh) or AR antagonists (Enza or ARN-509)[49]. Importantly, this study showed that androgen-deprivation in these mice resulted in reduced intra-tumoral androgen levels, leading to upregulation of androgen-repressed genes such as NOV. We next analyzed the RNA-seq data obtained from the Enza or ARN-509 treated mice (GSE95413), and a significant increase in the SPINK1 levels, along with other NEPC (SYP, CHGA, and TUBB3) and mesenchymal markers (VIM) was observed (Fig. 7a). Similar to transcriptomic data, a remarkable increase in the SPINK1 expression accompanied with NE and mesenchymal markers was observed by IHC in tumors of AR-antagonists treated mice, thus reaffirming the association between SPINK1 and NE-like phenotype (Fig. 7b, c). Intriguingly, an increase in the E-Cad (CDH1) expression was observed (Supplementary Fig. 9a, b), which is in line with recent contradictory report wherein E-Cad is shown to act as a survival-factor and supports metastases in mice model[50]. We also developed 22RV1 xenografts in immunodeficient mice (Crl:CD1-Foxn1[nu]) which were administered Enza after orchiectomy and evaluated the impact of androgen ablation in these xenografts. Similar to our VCaP xenografts' data, an increase in the expression of SPINK1

and NE-markers was noted in the 22RV1 tumors obtained from Enza-treated orchiectomized mice (Fig. 7d, e).

Since SPINK1 was found to be upregulated and associated with NE-markers in AR antagonists treated mouse xenografts, we next analyzed the RNA-seq data of the Beltran cohort[26] for SPINK1 expression. Interestingly, 8 out of 36 NEPC patients show increased expression of SPINK1 (Supplementary Fig. 9c). Next, to validate the expression of SPINK1, AR and NE-markers in these patients, we selected NEPC cases on the basis of SPINK1-high and SPINK1-low status, namely, WCM12, a patient who developed metastatic NEPC with liver metastases after treatment with ADT for metastatic prostate adenocarcinoma, and responded well to subsequent platinum-based chemotherapy[51]; WCM155, who also developed treatment-related NEPC after ADT with lung and liver metastases and responded well to the AURKA inhibitor, alisertib on a clinical trial[52]; and WCM677, who developed metastatic NEPC after treatment with ADT and subsequent radium for CRPC, and harbored somatic alterations in RB1, PTEN, and BRCA2[26]. Notably, similar to our SPINK1 and AR IHC data in prostate adenocarcinoma patients (Fig. 1), WCM12 also showed positive staining for SPINK1 and was negative for AR expression. WCM155 was developed as a patient-derived organoid which exhibited weak cytoplasmic staining for SPINK1 and was negative for AR expression. Conversely, WCM677 showed negative staining for SPINK1 expression and focal weak positive staining for AR (Fig. 7f). Collectively, our data demonstrate that androgen-deprivation using AR-antagonists leads to upregulation of SPINK1, which associates with NE-like features in our CRPC mice models. We also provide an important proof-of-concept highlighting the significance of SPINK1 in context of NEPC progression. However, these findings need to be interrogated using larger cohort, and an in-depth mechanistic study underpinning the role of SPINK1 in NEPC would provide further clarity.

## Discussion

SPINK1 expression in PCa has been associated with poor response to ADT, faster progression to castrate-resistant stage and cancer-associated mortalities[5,8,9], thus highlighting its significance as a biomarker of aggressivity and poor clinical response. A recent study showed that exogenous expression of HNF4G or HNF1A activates gastrointestinal-lineage transcriptome in PCa, and results in the upregulation of numerous PCa-gastrointestinal signature genes including SPINK1[53]. However, the exact mechanism of how SPINK1 is regulated in PCa, and why its upregulation is often associated with an aggressive phenotype remains unclear. Here, we provide compelling evidence that SPINK1 is an androgen-repressed gene, and that the use of AR antagonists relieve AR signaling-mediated repression of SPINK1 resulting in its upregulation. We also demonstrate that REST gets recruited to distinct RE-1 sites adjacent to the AR occupied AREs on the SPINK1 promoter in androgen-stimulated LNCaP cells, confirming its role as an AR transcriptional co-repressor in SPINK1 regulation. Furthermore, PCa specimens immunostained for SPINK1 and AR showed an inverse association, confirming AR-signaling mediated SPINK1 regulation. A recent study demonstrated that the anatomic location of the tumor in the prostate gland is influenced by AR signaling; tumors situated in the anterior lobe tends to have lower global AR signaling leading to differences in AR molecular subtypes, tumor size, and PSA[54]. Intriguingly, African-American men with aggressive PCa largely of SPINK1-positive subtype show higher propensity for anteriorly localized tumors as compared to Caucasian men with matched clinicopathologic features[54,55]. Previously, Paju et al. demonstrated reduced secretion of TATI

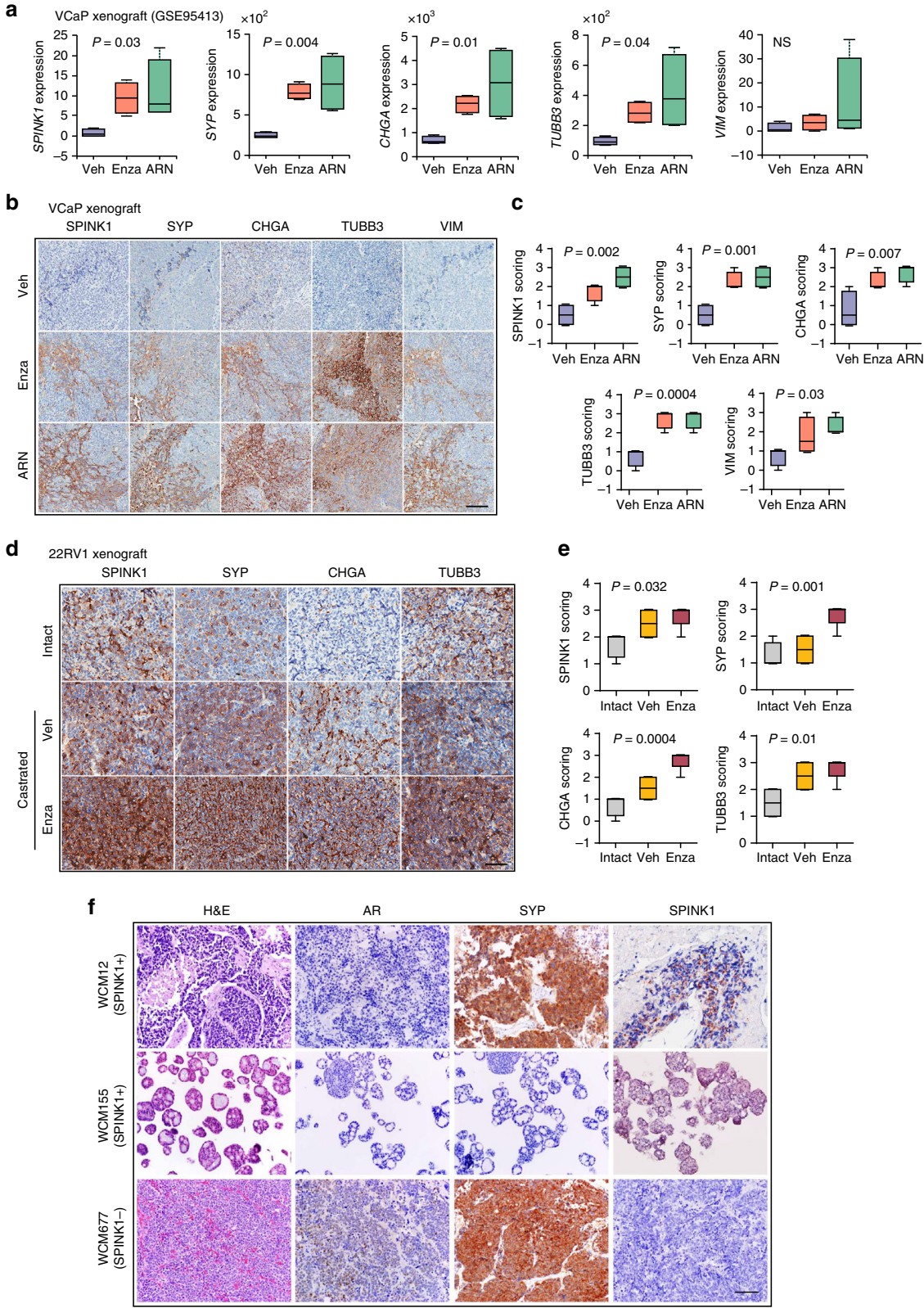

(SPINK1) in 22RV1 cells upon androgen stimulation and also showed its association with higher Gleason grade and expression of the neuroendocrine marker, CHGA[56]. Beltran and colleagues demonstrated absence of ERG oncoprotein in the NE foci of patients harboring *TMPRSS2-ERG* fusion, confirming the loss of androgen signaling in these patients[28]. Moreover, a recent report

indicated ~7% of the small cell neuroendocrine PCa patients were positive for SPINK1 expression[57]. Tumor multifocality remains a matter of concern in PCa molecular subtyping. Interestingly, SPINK1 expression is found to be restricted to few or more foci[58], we believe that low AR signaling in some of these foci (within anterior lobes) of prostate gland may results in SPINK1

**Fig. 7 ADT induced SPINK1 upregulation associates with NE-phenotype in mice and NEPC patients. a** Box plots depicting relative expression of *SPINK1*, *SYP*, *CHGA*, *TUBB3*, and *VIM* transcripts (read counts) in VCaP tumors implanted orthotopically in orchiectomized mice and subjected to vehicle ($n = 4$) or anti-androgens [enzalutamide ($n = 4$) or ARN-509 ($n = 4$)] treatment for 4 weeks (GSE95413). **b** Representative images of immunohistochemical staining for the same markers shown in **a** using VCaP xenograft tumors as described in **a**. Scale bar represents 100 μm. **c** Box plots depicting quantification of the immunohistochemical staining in VCaP xenografts for the markers shown in **b**. **d** Representative images for immunohistochemical staining of SPINK1, SYP, CHGA, and TUBB3 in 22RV1 xenograft tumors excised from orchiectomized mice treated with enzalutamide (20 mg/kg body weight) or vehicle control ($n = 5$ each). Intact group represents non-castrated control mice ($n = 5$). Scale bar represents 50 μm. **e** Box plots depicting quantification of the immunohistochemical staining in 22RV1 xenografts for the markers shown in **d**. **f** Representative images showing H&E staining (×200 magnification) and immunostaining (×200 magnification) for AR, synaptophysin, and SPINK1 in tumor specimens obtained from NEPC patients', namely WCM12, WCM155 (an organoid), and WCM677. Scale bar represents 100 μm. Data are presented as box-and-whisker plots with median, where the box extends from 25th–75th percentile, and whiskers ranges from minimum and maximum values. For panels **a**, **c**, **e** one-way ANOVA, Dunnett's multiple-comparisons test was applied.

overexpression (SPINK1-positive focus), which by acquiring additional critical alterations, such as loss of *RB1*, upregulation of *AURKA* and *MYCN* could drive NEPC.

The role of AR as a transcriptional activator in prostatic neoplastic progression has been well-established. Apart from its conventional tumor promoting activity, it has also been identified as a tumor suppressor[59,60]. Nevertheless, targeting androgen signaling by surgical or chemical castration remains the primary therapeutic modality for advanced stage PCa patients. Further, the mechanistic insights into androgen signaling led to the development of several AR-directed therapeutic interventions for men with castrate-resistant disease[61]. A co-repressor of AR, REST, also functions as a master negative regulator of the genes involved in neuronal differentiation by recruiting CoREST and SIN3A to the RE-1 elements of the target genes, which in turn recruits histone deacetylases (HDAC-1/2) and governs epigenetic reprogramming[62,63]. Alternatively, REST is shown to be a downstream effector of PI3K/AKT signaling, and inhibitors targeting this signaling axis results in enhanced degradation of REST through β-TRCP-mediated proteasome pathway, finally, resulting in NE-transdifferentiation[64]. Furthermore, FOXA1 recruits LSD1-CoREST complex and HDAC-1/2 to the androgen-regulated enhancers, and suppresses basal transcription of the target genes in AR-independent manner[65]. Based on these reports, we postulate that AR and REST repressive complex may also involve other co-factors such as CoREST in repressing the transcriptional regulation of *SPINK1*. Moreover, inactivation of REST in PCa cells show upregulation of neuronal specific genes[31]. Furthermore, REST negatively regulates EMT and stemness by repressing the expression of TWIST1 and CD44[66]. Importantly, our data revealed that targeting the ubiquitin-dependent REST degradation using iCK1 results in reduced expression of SPINK1 and REST targets, subsequently leading to decrease in oncogenic properties. Collectively, our findings suggest that AR and REST modulate the expression of *SPINK1*, thus stabilizing REST levels may be an alternate therapeutic strategy for controlling SPINK1-mediated oncogenicity and NEPC progression.

Ablation of androgen signaling has been implicated in upregulation of several EMT markers, a phenotype often associated with PCa metastases[16]. Further, a bidirectional negative-feedback loop between AR and ZEB1 has been established, which drives EMT and stem cell–like features upon androgen deprivation in LuCaP35 tumor explants[16]. Recently, we have shown that SPINK1 expression positively correlates with EZH2, a member of Polycomb repressive complex 2, known to induce pluripotency and stemness[13]. Furthermore, SOX2 has been implicated as a key regulator in governing pluripotency and drives NE-transdifferentiation[44]. Notably, knockdown of *Sox2* in mouse embryonic cells results in downregulation of *Spink3*, a mouse homolog of *SPINK1*[67]. Here, we demonstrated for the first time that suppression of SOX2 during NE-transdifferentiation in

LNCaP results in SPINK1 downregulation, while SOX2 overexpression upregulates SPINK1. Collectively, we propose a novel mechanism involved in SPINK1 regulation, whereby decreased AR and REST levels relieve the repression of the *SPINK1* promoter, and subsequently SOX2 gets recruited onto the *SPINK1* promoter to enhance its transcriptional activity.

Conclusively, our findings emphasize that administering PCa patients with AR targeted therapies may result in increased SPINK1 levels accompanied by upregulation of NE markers, potentially promoting the development of treatment-related NEPC (Fig. 8). Although, androgen ablation therapy is a well-established and highly effective treatment for PCa patients, resistance ultimately ensues. Understanding resistance mechanisms to ADT and subsequent AR therapies will eventually lead to more effective treatment strategies to improve outcomes for patients developing AR independent disease.

## Methods

**Human prostate cancer specimens**. Tissue microarrays (TMA) with prostate cancer (PCa) specimens were obtained from the Department of Pathology, Henry Ford Health System (HFHS), Detroit, Michigan, USA and Vancouver Prostate Centre (VPC), Vancouver, BC, Canada, after acquiring the due consent from the patients and mandatory approval from the Institutional Review Board. The HFHS TMA comprises radical prostatectomy cases, mostly with localized cancer and some with lymph node metastases. The VPC TMA comprises of hormone naive cases ($n = 33$) and cases administered with neoadjuvant hormone therapy ($n = 55$), comprising LHRH agonists and bicalutamide for 3 months. The TMAs were stained for SPINK1 and AR using immunohistochemistry (IHC). NEPC patients' specimens namely, WCM12, WCM155 and WCM677 were obtained from the Department of Pathology and Laboratory Medicine, Weill Cornell Medicine, New York, USA after getting approval from the institutional review board[26,51,52], and subjected to haematoxylin and eosin (H&E) and IHC for AR, SPINK1 and SYP. All patients' specimens used in this study were collected in accordance with the ethical principles founded in the Declaration of Helsinki.

**Mice xenograft studies**. The immunodeficient mice (Crl:CD1-*Foxn1^{nu}*) were initially procured from the Charles River Laboratory and the breeding colonies were maintained in specific-pathogen-free facility as per the guidelines. For mice xenograft studies, 5-6 weeks old male immunodeficient mice were anaesthetized using ketamine (50 mg/kg) and xylazine (5 mg/kg) and were subcutaneously implanted with 22RV1 cells ($3 \times 10^6$) resuspended in 100 μl of saline with 30% Matrigel in the dorsal flanks on both the sides. Once the tumor burden reached average ~150 mm³, mice were randomized into 3 groups ($n = 5$ each). One group was kept as intact with no surgical procedure or treatment. In other two groups, orchiectomy procedure was performed and were orally administered either vehicle control [5% dimethyl sulfoxide (DMSO), 30% polyethylene glycol 400 (PEG-400), 65% corn oil] or enzalutamide (MedChemExpress, HY-70002; 20 mg/kg body weight) dissolved in vehicle control, five times a week. The drug treatment was continued for two weeks, subsequently the mice were euthanized, and the tumors were excised, fixed in 10% neutral buffered formalin, paraffin embedded and subjected to immunohistochemical staining for various markers. All procedures implemented in this study were approved by the Committee for the Purpose of Control and Supervision of Experiments on Animals (CPCSEA), Ministry of Environment, Forest and Climate Change, Govt. of India and conform to the regulatory standards of the Institutional Animal Ethics Committee of the Indian Institute of Technology Kanpur and CSIR-Central Drug Research Institute, Lucknow, India.

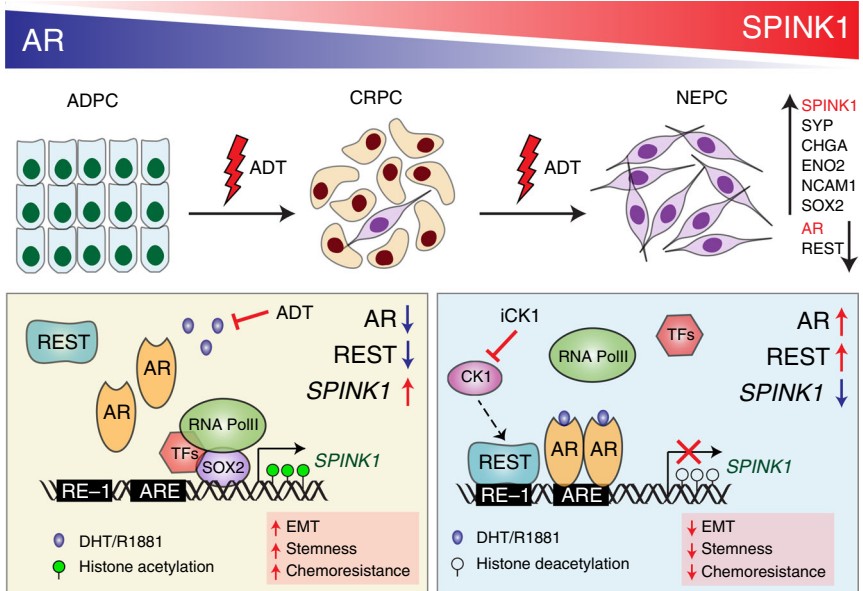

**Fig. 8 Androgen receptor and its transcriptional co-repressor REST modulate *SPINK1* expression.** Schematic showing androgen deprivation therapy (ADT) induced progression of prostate adenocarcinoma (ADPC) to neuroendocrine prostate cancer (NEPC) (top panel). ADT using AR-antagonists relieve AR-REST mediated transcriptional repression of *SPINK1*, and subsequently, SOX2 gets recruited on the *SPINK1* promoter resulting in its upregulation (left box). In addition, Casein kinase 1 inhibitor (iCK1) restores REST levels causing downregulation of SPINK1, leading to reduced stemness and cellular plasticity (right box).

**Immunohistochemistry (IHC) staining**. IHC for AR and SPINK1 was performed using EnVision FLEX system (Agilent). Briefly, TMAs and tumor xenograft slides were incubated at 60 °C for 2 h and antigen retrieval was carried out in EnVision FLEX Target Retrieval Solution, High pH (Agilent DAKO, K800421-2) in a PT Link instrument (Agilent DAKO, PT200). Slides were washed in 1× EnVision FLEX Wash Buffer (Agilent DAKO, K800721-2) for 5 min, followed by treatment with Peroxidazed 1 (Biocare Medical, PX968M) for 5 min, followed by Background Punisher (Biocare Medical, BP974L) for 10 min with a wash after each step. Mouse monoclonal SPINK1 (Novus Biologicals, H00006690-M01, 1:100 dilution), AR (CST, 5153, 1:200 dilution), SYP (Novocastra, NCL-SYNAP299, 1:200), CHGA (BioGenex, LK2H10, 1:400), TUBB3 (Abcam, ab68193, 1:250), VIM (Dako, M7020, 1:500) and ECAD (Dako, M3612, 1:200) antibodies diluted in EnVision FLEX Antibody Diluent (Agilent DAKO, K800621-2) were added onto each slide and incubated overnight at 4 °C. Slides were washed and incubated in Mach2 Doublestain 1 (Biocare Medical, MRCT523L) for 30 min at room temperature. Subsequently, slides were rinsed in 1× EnVision Wash Buffer thrice, and treated with a Ferangi Blue solution (Biocare Medical, FB813S) for 7 min. Slides were rinsed twice in distilled water and stained with EnVision FLEX Hematoxylin (Agilent DAKO, K800821-2) for 5 min. After multiple washes, slides were immersed in a 0.01% ammonium hydroxide solution and rinsed twice in distilled water. Once the slides were dried completely, they were put in xylene for approximately for 5 min and mounted using EcoMount (Biocare Medical, EM897L).

**Evaluation criteria for IHC staining**. The IHC staining for SPINK1 was evaluated as positive or negative as described previously[68]. The staining for AR was scored into four different categories: high, medium, low, and negative, based upon staining intensity. Further, association between SPINK1 and AR expression in patients' samples was inferred by applying Chi-square test and Fisher's exact test on GraphPad Prism 7. For both the xenograft models, 6 random fields from each tumor tissue were selected and IHC scoring was divided into four categories: 0 (no staining), 1 (weak staining), 2 (moderate staining), and 3 (strong staining) based on intensity.

**AR signaling score**. AR signaling score was obtained using previously published AR gene signature[43,69]. Firstly, PCa patients from TCGA-PRAD and MSKCC cohorts were stratified by performing quartile-based normalization to classify the patients based on high and low *SPINK1* expression. For the signaling score, gene expression of each gene mentioned in the AR gene signature was downloaded as respective *Z*-Score from cBioPortal and the score was then determined by computing the mean expression of these genes. For both the cohorts, signaling score between the top and bottom quartile was compared. Significance of the signaling score among *SPINK1*-high and *SPINK1*-low patients was evaluated by unpaired two-tailed Student's *t* test on GraphPad Prism 7.

**Analysis of TCGA-PRAD dataset**. Illumina HiSeq mRNA data of patients with prostate adenocarcinoma (PRAD) was downloaded from TCGA dataset available at the UCSC Xena (https://xena.ucsc.edu/) for *SPINK1*, *AR*, *NCOR1*, *NCOR2*, *NRIP1*, *SYP*, and *REST* genes. Since, *SPINK1* gene is overexpressed in ~10-15% of the PCa patients[5], we performed quartile based normalization[34] to stratify the patients based on high and low *SPINK1* expression. Accordingly, patients corresponding in the top quartile ($n = 125$) (QU, $\log_2$ (RPM + 1) > 5.468 or $\log_2$ (normalized count + 1) > 1.892) were considered as *SPINK1*-high whereas patients in the lower quartile ($n = 125$) (QL, $\log_2$ (RPM + 1) < 1.124 or $\log_2$ (normalized count + 1) < −2.611) were assigned as *SPINK1*-low. The corresponding gene expression values for *AR*, *NCOR1*, *NCOR2*, *NRIP1*, *SYP*, and *REST* in *SPINK1*-high versus *SPINK1*-low patients were compared to identify the association between *SPINK1* and these genes. For the heatmap between *AR* and *SPINK1*, PCa patients from TCGA-PRAD cohort were grouped based on high and low *AR* expression ($\log_2$ (RPM + 1) > 9.4 and $\log_2$ (RPM + 1) < 8.1). The high and low cutoff value for *AR* expression was obtained from the TCGA portal (https://xenabrowser.net). The corresponding *SPINK1* expression values in *AR*-high and *AR*-low patients was further used to construct the heatmap by employing gplot function in R. Correlation plots between *SPINK1*, *SYP*, *MYCN*, *CHGB*, *REST* and its complex members derived from the MSKCC cohort were directly retrieved from cBioPortal (http://www.cbioportal.org).

**Cell lines and authentication**. All the prostate cancer (22RV1, VCaP, LNCaP, PC3, and DU145), benign prostatic epithelial (RWPE-1), colorectal cancer (WiDr), pancreatic cancer (CAPAN-1), melanoma (SK-MEL-173) and human embryonic kidney 293 (HEK293FT) cell lines were obtained from American Type Cell Culture (ATCC) and maintained as per guidelines. CWR22Pc cells were a kind gift from Dr. Marja T. Nevalainen[70]. C4-2 cells were a generous gift from Dr. Mohammad Asim and were cultured according to ATCC guidelines. 16D^CRPC, 42D^ENZR, and 42F^ENZR cells were kindly gifted by Dr. Amina Zoubeidi[29]. Briefly, cells were cultured in the recommended media supplemented with 10% fetal bovine serum (FBS) (Gibco) and 0.5% Penicillin Streptomycin (Pen Strep) (Thermo Fisher Scientific) in cell culture incubator (Thermo Fisher Scientific) supplied with 5% $CO_2$ at 37 °C. Cell line authentication was done via short tandem repeat (STR) profiling at the Lifecode Technologies Private Limited, Bangalore and DNA Forensics Laboratory, New Delhi. Routine *Mycoplasma* contamination of was checked using PlasmoTest mycoplasma detection kit (InvivoGen).

**Plasmids and constructs**. The pGL3-SPINK1-PP construct was obtained by cloning the *SPINK1* proximal promoter (SPINK1-PP) in pGL3-basic vector, a kind gift from Dr. Amitabha Bandyopadhyay. pGL3-SPINK1-DP was generated by cloning distal promoter of the *SPINK1* gene in pGL3-SV40 enhancer vector (Promega). Site-directed mutagenesis was performed to alter the androgen response element (ARE) in the *SPINK1* distal promoter (SPINK1-DP) to create the pGL3-SPINK1-DP mutant (MT) from pGL3-SPINK1-DP wildtype (WT). The

pcDNA3.1(+) SOX2 was purchased from GenScript, pcDNA3.1(+)-SPINK1-2xV5 was synthesized from GeneArt and pcDNA3.1(+) empty vector was kindly gifted by Dr. Arun K. Shukla. Wild type and mutant (ΔNLS and V581F) AR constructs cloned in FUCGW lentiviral vectors were generously provided by Dr. Owen Witte. pGIPZ plasmids (shScrambled, shSPINK1-1, shSPINK1-2 and shSPINK1-3) were procured from Dharmacon. Lentiviral overexpression constructs pLV-SPINK1, pLV-REST and control pLV vectors were purchased from VectorBuilder. pLKO.1 (shScrambled and shREST) were procured from Dr. Subba Rao Gangi Setty. Doxycycline-inducible AR-V7 overexpression plasmid (pHAGE-ARV7) was a kind gift from Dr. Nancy L. Weigel. pGL4.10-PSA construct was a generous gift from Dr. Jindan Yu.

**Lentiviral packaging**. Lentiviral particles were produced using ViraPower Lentiviral Packaging Mix (Invitrogen) according to the manufacturer's instructions. Briefly, HEK293FT were transfected with the plasmids of the packaging mix along with shRNA/overexpressing constructs. The viral particles were harvested 60–72 h post-transfection and stored at −80 °C. For generating stable lines, cells were infected with lentiviral particles along with polybrene (hexadimethrine bromide; 8 μg/ml) (Sigma-Aldrich). Culture media was changed next day and puromycin (Sigma-Aldrich, R0908) selection was started three days of post-infection.

**Androgen stimulation and deprivation**. For androgen stimulation, cells were starved for 72 h in phenol-red free media (Gibco) supplemented with 5% charcoal stripped serum (CSS) (Gibco) followed by stimulation with R1881 (Sigma-Aldrich) at the indicated time points. For anti-androgen treatment, VCaP cells were serum-starved for 6–8 h using DMEM media supplemented with GlutaMAX (Gibco) followed by treatment with different concentrations of enzalutamide (MedChemExpress, HY-70002) and bicalutamide (Sigma-Aldrich, B9061) for 48 h in complete media. For long-term androgen deprivation, LNCaP, LNCaP AR-V7 with or without doxycycline induction (40 ng/ml), LNCaP-shSCRM and LNCaP-shSPINK1 cells were cultured in RPMI-1640 phenol-red free medium supplemented with 5% CSS.

**Enzyme-linked immunosorbent assay (ELISA)**. SPINK1 protein level in total cell lysate (CL) and conditioned media (CM) was quantified using Human SPINK1 ELISA kit (R&D Systems, DY7496) as per the manufacturer's instructions. Briefly, CM was prepared by culturing cells in the respective media containing 5% charcoal stripped serum (CSS). After R1881 stimulation or anti-androgen treatment, the media was collected, centrifuged at $461 \times g$ for 10 min at 4 °C to remove cellular debris and supernatant was used for the ELISA-based quantification.

**Transient transfection**. 22RV1 and VCaP cells were plated at 40–45% confluency and transfected with 30pmol of small interfering RNA (siRNA) against *AR* (Dharmacon, Cat No. LU-003400-00-0002), *SPINK1* (Dharmacon, Cat. No. LU-019724-00-0002), *SOX2* (Thermo Fisher Scientific, Cat No. 4392420) and non-targeting (NT) control (Dharmacon, Cat. No. D−001810-10-05) using Lipofectamine RNAiMAX Transfection Reagent (Thermo-Fisher Scientific) according to manufacturer's instructions.

**Real time quantitative PCR**. Total RNA was extracted using TRIzol (Ambion) and 1 μg of total RNA was used for cDNA synthesis using SuperScript III First-Strand Synthesis System (Invitrogen) according to the manufacturer's instructions. For qPCR, all reactions were performed in triplicates using SYBR Green PCR Master-Mix (Applied Biosystems). Relative target gene expression was calculated for each sample using the ΔΔCt method[11], using primers mentioned in the Supplementary Table 2.

**Gene expression analysis**. For gene expression profiling, the total RNA from 22RV1 cells (shSCRM, shSPINK1-1, shSPINK1-2 and shSPINK1-3) was collected and subjected to Agilent Whole Human Genome Oligo Microarray profiling (dual color) according to manufacturer's protocol using Agilent platform (8 × 60 K format). Three separate microarray hybridizations were performed, using 22RV1-shSPINK1 cells against control 22RV1-shSCRM cells. For the microarray data Lowess (locally weighted regression) normalization was performed. The gene expression pattern for differentially regulated genes was identified using hierarchical clustering implemented Pearson coefficient correlation algorithm. Benjamini and Hochberg procedure was used to calculate FDR-corrected *P*-values (with FDR < 0.05) to identify differentially expressed genes. Our analysis revealed 697 genes downregulated in 22RV1-shSPINK1 cells (log₂ fold change > 0.5 or < −0.5, 90% confidence interval), which were further analyzed for enriched pathways (*P* < 0.05) using DAVID (Database for Annotation, Visualization and Integrated Discovery). Gene set enrichment analysis (GSEA) was performed to identify gene-sets enriched in androgen deprived LNCaP cells (GSE8702). RNA-sequencing data of 42D^ENZR and 16D^CRPC cells was obtained from Dr. Amina Zoubeidi's laboratory[29]. Heatmap.2 function of gplots in R was used to generate heatmaps for the publicly available datasets wherein 22RV1 (GSE71797) and VCaP cells (GSE51872) were stimulated with R1881 and dihydrotestosterone (DHT), respectively. RNA-seq dataset, GSE80743 was analyzed for *SPINK1* expression in AR splice variants-

silenced 22RV1 cells, and GSE95413 data of the VCaP tumors excised from anti-androgens treated CRPC mice xenograft model was analyzed for the expression of SPINK1, NEPC and EMT markers.

**Western blot analysis**. Cell lysates were prepared in radioimmunoprecipitation assay (RIPA) lysis buffer, supplemented with complete Protease Inhibitor Cocktail (Roche) and Phosphatase Inhibitor Cocktail Set-II (Calbiochem). Protein samples were resolved on the SDS-PAGE and transferred onto a polyvinylidene difluoride (PVDF) membrane (GE Healthcare). The membrane was blocked with 5% non−fat dry milk in tris-buffered saline, 0.1% Tween 20 (TBS-T) for 1 h at room temperature, and then incubated overnight at 4 °C with the following primary antibodies: 1:500 diluted SPINK1 (R&D Systems, MAB7496-SP), 1:1000 diluted AR (CST, 5153), 1:2000 diluted REST (Abcam, ab75785), 1:2000 diluted PSA (CST, 5877), 1:1000 diluted E-Cadherin (CST, 3195), 1:1000 diluted N-Cadherin (Abcam, ab98952), 1:1000 diluted Vimentin (abcam, ab92547), 1:1000 diluted phospho-Akt (CST, 13038), 1:1000 diluted total-Akt (CST, 9272), 1:1000 diluted Histone H3 (CST, 14269), 1:2000 diluted SYP (Abcam, ab32127), 1:1000 diluted ENO2 (Abcam, ab53025), 1:1000 diluted SOX2 (Abcam, ab97959) and 1:5000 diluted β-actin (Abcam, ab6276). Subsequently, blots were washed in 1X TBS-T buffer and incubated with respective horseradish peroxidase-conjugated secondary anti-mouse or anti-rabbit antibody (Jackson ImmunoResearch) for 2 h at room temperature. After washing, the signals were visualized by enhanced chemiluminescence system as described by the manufacturer (GE Healthcare). For all immunoblot experiments, β-actin was used as a loading control.

**Immunofluorescence staining**. Cells were grown on glass coverslips in 24-well culture dishes and fixed with 4% paraformaldehyde. The cells were washed with 1× PBS, permeabilized with 0.3% Triton X-100 in 1× PBS (PBS-T) for 10 min and blocked using 5% normal goat serum in PBS-T for 2 h at room temperature. The cells were then incubated with primary antibodies: SPINK1 (1:100, Abnova H00006690-M01), AR (1:200, CST 5153), ERG (1:200, ab92513), E-Cadherin (1:400, CST 3195), Vimentin (1:100, CST 3932) diluted in PBS-T, at 4 °C overnight. Cells were washed using 0.05% Tween 20 in 1× PBS, followed by incubation with Alexa Fluor conjugated anti-mouse or anti-rabbit secondary antibodies (1:600 dilution, CST #4408 and #4412, respectively). After washing with PBS-T, cells were stained with TRITC-Phalloidin (Sigma-Aldrich) followed by DAPI (Sigma-Aldrich). The coverslips were mounted on glass slides using Vectashield mounting medium (Vector laboratories). Images were captured using Axio Observer Z1 inverted fluorescence microscope (Carl Zeiss) equipped with an Apotome device. For quantification, the images were captured with ×63 oil immersion objective (NA 1.4). Post-processing of the acquired images was done using ImageJ software. The boundary of each cell was marked and cells with area ranging between 70 and 130 μm² were considered to quantify mean fluorescence intensity per unit area. Statistical significance was calculated using one-way ANOVA or *t*-test depending on the number of groups in the datasets. For all immunostaining experiments, F-actin and nucleus was stained by TRITC-phalloidin and DAPI, respectively. For neurite outgrowth quantification, 20 random fields were imaged for stable LNCaP-AI-shSPINK1 and LNCaP-AI-shSCRM green fluorescent protein (GFP) positive cells cultured in the androgen deprived condition for a duration of 30 days. The neurite lengths were measured by using the Simple Neurite Tracer (http://imagej.net/Simple_Neurite_Tracer).

**Chromatin immunoprecipitation (ChIP) assay**. Cells were crosslinked with 1% formaldehyde for 10 min followed by quenching with 125 mM Glycine for 5–8 min. The cells were washed twice with phosphate buffered saline (1× PBS) and lysed with the lysis buffer [1% SDS, 50 mM Tris-Cl (pH 8.0), 10 mM EDTA, protease inhibitor cocktail (Roche) and phosphatase inhibitor cocktail set-II (Calbiochem)]. The cell lysate was then sonicated using Bioruptor (Diagenode) to obtain ~500 bp DNA fragments. The sheared chromatin was collected after centrifugation and incubated overnight at 4 °C with 4 μg of either primary or isotype control antibodies. ChIP was carried out using the following antibodies: AR (CST, 5153), Rpb1 CTD (CST, 2629), H3K9Ac (CST, 9649), REST (Abcam, ab70300), SOX2 (Abcam, ab97959), Histone H3 (CST, 14269), Phospho-Rpb1 CTD (Ser5) (CST, 13523), Phospho-Rpb1 CTD (Ser2) (CST, 13499), H3K27me3 (CST, 9733), H3K4me3 (CST, 9751) and isotype control antibodies, rabbit IgG (Invitrogen, 10500C) and mouse IgG (Invitrogen, 10400C). Simultaneously, Dynabeads coated with Protein G (Invitrogen) were blocked using 500 μg/ml of sheared salmon sperm DNA (Sigma-Aldrich) and 100 μg/ml bovine serum albumin (BSA) (HiMedia) overnight at 4 °C. Blocked beads were washed using dilution buffer [1% Triton X-100, 2 mM EDTA, 150 mM NaCl, 20 mM Tris-Cl (pH 8.0) with protease inhibitor cocktail (Roche) and phosphatase inhibitor cocktail set-II] and incubated for 6–8 h at 4 °C with the lysate containing antibody to make antibody-bead conjugates. Next, the beads conjugated with antibody were washed thrice in low salt wash buffer (1% Triton X-100, 150 mM NaCl, 0.1% SDS, 20 mM Tris-HCl (pH 8.0), 2 mM EDTA with protease inhibitor cocktail and phosphatase inhibitor cocktail set-II) and once with high salt wash buffer (same as previous wash buffer except 500 mM NaCl) followed by a final wash with 1× TE buffer. The immunocomplex was eluted using elution buffer [1% SDS, 100 mM NaHCO₃, Proteinase K (Sigma-Aldrich) and RNase A (500 μg/ml each) (Sigma-Aldrich)]. DNA was isolated using phenol-

chloroform-isoamyl alcohol extraction. Precipitated DNA was washed with 70% ethanol, air dried and resuspended in nuclease free water (Ambion). The ChIP-qPCR was performed using primers mentioned in the Supplementary Table 2.

**Chromatin immunoprecipitation sequencing (ChIP-Seq) analysis**. To determine the recruitment of AR on *SPINK1* promoter, we analyzed publicly available ChIP-Seq data (GSE58428) for AR in VCaP cells treated with DHT and vehicle control, ethanol (EtOH). Raw single-end reads were analyzed for its quality using FASTQC, followed by trimming with FASTQ Trimmer, ensuing all the default settings of Galaxy web platform available on the public server at usegalaxy.org. Reads were aligned to the reference genome (hg18) using Bowtie to generate Sequence Alignment/Map (SAM) files. Unaligned or unmapped reads were filtered using FilterSAM, a utility of SAMtools. The SAM files were converted to its Binary Alignment/Maps (BAM) files using SAMtools. Further, ChIP-Seq peaks for DHT and EtOH treated samples were called using Model-based analysis of ChIP-seq (MACS; $P < 10^{-5}$) with default settings against Input. BAM and BED files obtained were visualized by Integrative Genomic Browser (IGB).

**Luciferase promoter reporter assay**. 22RV1, VCaP or SOX2 overexpressing LNCaP cells at 40–50% confluency in a 24-well plate were transfected with pGL3-SPINK1-PP (250 ng) and pRL-null vector (2.5 ng) using FuGENE HD Transfection Reagent (Promega). For androgen stimulation, 22RV1 cells were serum starved for 8 h and stimulated with R1881 at indicated concentrations for 18 h in RPMI-1640-PRF media containing 5% charcoal stripped serum (CSS). For anti-androgen treatment, VCaP cells were treated with enzalutamide (10 μM) for 24 h. After 48 h of transfection with luciferase constructs, cells were harvested using the lysis buffer provided with Dual-Glo Luciferase assay kit (Promega). Firefly and Renilla luciferase activity were measured according to the manufacturer's protocol using GloMax® 96 Microplate Luminometer (Promega). For each sample, firefly luciferase activity was normalized to Renilla luciferase activity. Same protocol was followed to measure the luciferase promoter reporter activity for the pGL3-SPINK1-DP wildtype, pGL3-SPINK1-DP mutant and pGL4.10-PSA constructs.

**Migration assay**. Migration assay was performed using Transwell Boyden chamber of 8 μm pore size (Corning). Briefly, after the desired treatment, cells were trypsinized, counted and $1 \times 10^5$ cells were resuspended in serum-free media and added to the upper chamber of the Transwell. The bottom chamber was supplemented with cell culture media supplemented with 30% FBS. After 48 h of incubation, the migrated cells were fixed with 4% paraformaldehyde and stained with 0.5% (w/v) crystal violet. Representative pictures were captured using Axio Observer Z1 microscope (Carl Zeiss). For quantification, the Transwells were destained using 10% acetic acid (v/v) in distilled water. Absorbance was measured at 550 nm.

**Chemosensitivity assay**. For determining the $IC_{50}$ of drugs, 22RV1-shSCRM and -shSPINK1-1 cells ($3 \times 10^3$) were plated in 96-well dishes and treated with varying concentration of drugs for 48 h. The $IC_{50}$ of the drugs was determined using WST-1 (Roche) as per manufacturer's instructions.

**Casein kinase 1 inhibitor (iCK1) treatment**. 22RV1 cells were serum starved for 12 h in RPMI-1640 medium (Gibco) and then treated with different concentrations of iCK1, D4476 (MedChemExpress, HY-10324) for 60 h in complete media.

**Prostatosphere assay**. 22RV1-shSCRM and -shSPINK1 cells ($1 \times 10^4$) were plated in low adherence 6-well cell culture dishes in serum-free DMEM-F12 media (1:1, Invitrogen) supplemented with B27 (1×, Invitrogen), EGF (20 ng/ml, Invitrogen), FGF (20 ng/ml, Invitrogen) as previously described[13]. A small population of cells forming prostatospheres were collected by centrifugation and were mechanically dissociated into single cell suspension, followed by re-plating in fresh culture media. Next, the new prostatospheres formed were again passaged in the similar manner for multiple generations (every 3rd day), and the experiment was terminated after two weeks. The prostatospheres formed were assessed for sphere forming efficiency. Mean area of spheres was measured using ImageJ software, and the spheres >50 μm in diameter were counted and the values were represented as percent sphere formation efficiency. For three-dimensional tumor spheroid assay, 22RV1 cells ($3 \times 10^3$) were resuspended in complete media supplemented with 2% growth factor reduced (GFR) Matrigel and plated in the chamber slide precoated with GFR Matrigel. Complete media supplemented with Casein Kinase 1 inhibitor, iCK1 (20 μM) or DMSO control was changed every two days for 2 weeks. The tumor spheroids of size >1000 μm in diameter were counted and the values were represented as percent sphere formation efficiency and the mean area of spheres were plotted.

**Flow cytometry experiments and analysis**. For Hoechst side population (SP) Assay, 22RV1-shSCRM and -shSPINK1 cells ($5 \times 10^5$) were stained with Hoechst 33342 (5 μg/ml) in the presence or absence of verapamil (100 μM) (Sigma-Aldrich), ATP-binding cassette transporter (ABC transporter) inhibitor (Thermo Fisher Scientific) with agitation for 2 h at 37 °C. Cells were then washed and resuspended

in FACS buffer (1× PBS, 2% fetal bovine serum and 0.1% sodium azide), followed by Propidium iodide (PI) (5 μg/ml) staining (BioLegend) to exclude the dead cells. The side population (SP) was detected using UV laser at 350 nm. The Hoechst blue and far-red fluorescence was measured using 460/50 and 670/30 long-pass filter, respectively. Higher laser power was used to capture red channel as its intensity is lower than blue channel. Optimal resolution of SP cells was obtained by using a laser power of 30–35 mW. For gating of side population tail, cells were first gated for the live population using forward scatter (FSC) and side scatter (SSC) dot plot; further a dim tail of SP were then gated and displayed as a dot plot for Hoechst blue (460/50) and far red (670/30) scatter. For each condition, ~$1 \times 10^5$ events were acquired on BD Influx™ Cell Sorter. Data analysis and acquisition was done using the FlowJo version 10.7.

For aldehyde dehydrogenase (ALDH) activity, Aldefluor assay was performed using Aldefluor kit (Stem Cell Technologies) following the manufacturer's guidelines. Briefly, cells were washed with 1× PBS and centrifuged at $260 \times g$ for 5 min at 4 °C, and were resuspended in 1 ml Aldefluor assay buffer, followed by addition of 5 μl of activated Aldefluor substrate. Cell suspension was immediately divided in 2 parts, one tube with 5 μl of ALDH inhibitor, diethylaminobenzaldehyde (DEAB) and the other tube without DEAB. Cells were incubated at 37 °C for 30 min, centrifuged and resuspended in 500 μl of Aldefluor assay buffer. The Aldefluor activity was detected in green (FITC) channel. For gating ALDH-positive cell population, cells were first gated for live-cell population using forward scatter (FSC) and side scatter (SSC) dot plot. Further, a FITC channel versus SSC dot plot was created and DEAB treated cells were used to gate the control population. The same gate was applied over corresponding sample tube containing ALDH to demarcate the ALDH-positive population. For each condition, ~$1 \times 10^5$ events were acquired on BD FACSCanto II (BD Biosciences) or Beckman Coulter's CytoFLEX platform and the analysis was performed using FlowJo version 10.7.

For cell cycle analysis, *SPINK1*-silenced 22RV1 cells were fixed with 70% ethanol and stained with PI using manufacture's protocol, and analyzed using in-built univariate model of FlowJo version 10.7. For apoptosis assay, same cells were washed with cold 1× PBS and resuspended in 1X binding buffer ($1 \times 10^6$ cells/ml). Subsequently, staining was performed using PE Annexin V Apoptosis Detection Kit I (BD Pharmingen), following the manufacturer's instructions. Quadrants defining cells undergoing apoptosis were gated on Annexin V (PE) versus 7AAD (PerCP) dot plots using unstained, Annexin V (PE) and 7AAD (PerCP) single stained cells. The quadrants were defined as Annexin$^+$/7AAD$^-$ (early-apoptotic), Annexin$^-$/7AAD$^+$ (necrotic) and Annexin$^+$/7AAD$^+$ (late-apoptotic) cells. For each condition, ~$1 \times 10^4$ events were acquired on Beckman Coulter's CytoFLEX platform and analyzed using FlowJo version 10.7.

For cell-surface marker staining, cells were stained with anti-human CD56 (NCAM1, PE/Cy7) antibody (eBioscience, Ab 25-0567-42, 1:50) and CD117-APC (c-KIT) antibody (Miltenyi Biotec, 130-098-207, 1:40) followed by 1 h incubation at 4 °C. Cells were first gated for live-cell population using forward scatter (FSC) and side scatter (SSC) dot plot. Histograms of specific antibody-stained cells were generated and compared to respective isotype controls. For each condition, ~$1 \times 10^5$ events were acquired on BD FACSCanto II (BD Biosciences) or Beckman Coulter's CytoFLEX platform and analyzed with FlowJo version 10.7.

**Cell proliferation assay**. Cell proliferation assay was carried out by plating 3000 cells per well in 96-well plate. Cells were treated with different concentrations of Casein Kinase 1 inhibitor (iCK1) against DMSO control and incubated for the indicated time points. Cell viability was determined following incubation with the Cell Proliferation Reagent WST-1 (Roche), followed by colorimetric assay as per manufacturer's protocol.

**Foci formation assay**. 22RV1 cells ($2 \times 10^3$) were plated in six-well culture dishes in RPMI-1640 medium (Gibco) supplemented with 5% fetal bovine serum (Invitrogen) and incubated at 37 °C, media was changed after every 48 h with Casein Kinase 1 inhibitor, iCK1 (20 μM) along with DMSO control. The assay was terminated after 2 weeks and foci were fixed with 4% paraformaldehyde and stained with crystal violet solution (0.1% w/v). For destaining, 10% glacial acetic acid was used, and the absorption was quantified at 550 nm.

**Statistical analysis**. Statistical significance was determined using either one-way ANOVA, two-way ANOVA with post hoc multiple comparisons test or unpaired two-tailed Student's *t*-test for most experiments or otherwise mentioned. The differences between the different groups were considered significant if the *P*-value was less than 0.05. Significance is indicated as follows: *$P < 0.05$, **$P < 0.001$ and NS denotes not significant. Error bars represent standard error of the mean (SEM) obtained from experiments performed at least three independent times.

**Reporting summary**. Further information on research design is available in the Nature Research Reporting Summary linked to this Article.

## Data availability
The gene expression microarray data for SPINK1-silenced 22RV1 cells generated in this study has been submitted to the NCBI Gene Expression Omnibus under the accession

number GSE124345. There are various other datasets used in the study, namely: ChIP-Seq dataset for AR binding in androgen stimulated VCaP cells, GSE58428, Microarray dataset for long-term androgen deprivation of LNCaP cells, GSE8702, Microarray dataset of VCaP cells with and without androgen (DHT) stimulation, GSE51872, RNA-Seq dataset of 22RV1 cells with and without androgen (R1881) stimulation, GSE71797, RNA-Seq dataset for AR splice variants-silenced 22RV1 cells, GSE80743, RNA-seq dataset of vehicle, enzalutamide and ARN-509 treated castrate-resistant VCaP tumors after 4-weeks of treatment, GSE95413. The databases used in this study include: cBioPortal and UCSC Xena for analyzing the correlation plots and downloading gene expression values from MSKCC and TCGA-PRAD cohorts. The source data underlying Figs. 2a, d, h, l, m, 4a, 5d, h, l, m, 6e, f, g, k and Supplementary Figs. 2b, l, 3e, h, i, 4e, 7a, 8f, g, j, k for gel images have been provided as Source Data file.

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

## Acknowledgements

B.A. is an Intermediate Fellow of the Wellcome Trust/DBT India Alliance. This work is supported by the Wellcome Trust/DBT India Alliance Fellowship [grant number: IA/I (S)/12/2/500635] awarded to B.A. Financial support from Department of Biotechnology (BT/PR8675/GET/119/1/2015; to B.A.) and Science and Engineering Research Board (EMR/2016/005273; to B.A.), Government of India, is also acknowledged. N.M. acknowledges fellowship support from University Grants Commission (UGC), Government of India. R.T. thanks the Indo-Canadian Shastri Research Student Fellowship (SRSF) for short-term travel support. N.P. thanks the Department of Defense for the grant support (CDMRP W81XWH-16-1-0544). D.D. acknowledges financial support from DST (EMR/2016/006935) and DBT (BT/AIR0568/PACE-15/18) and CSIR fellowship grant to M.A.N. We are grateful to Prof. Noel Buckley for his insightful suggestions. We thank Anjali Bajpai for critically reading the paper. This work is supported by the Wellcome Trust/ DBT India Alliance grant (IA/I(S)/12/2/500635 to B.A.).

## Author contributions

R.T., N.M. and B.A. designed and directed the experimental studies. R.T., N.M., and B.A. performed in vitro and in vivo studies. R.T., N.M. and V.B. performed the gene expression studies, bioinformatics analysis, ChIP assays and flow cytometry experiments. R.T. and A.Y. performed the immunofluorescence. M.A.N. and D.D. assisted in executing mice xenograft studies. M.P. provided the VCaP xenograft tumor specimens. A.Z. provided the neoadjuvant hormone therapy tissue microarray data and LNCaP xenograft-derived cell lines. H.B., M.S. and F.K. analyzed and interpreted the RNA-Seq data of Beltran cohort, provided NEPC patients' specimens and related clinical information. S.C., N.G. and N.P. assembled PCa tissue microarrays and performed immunohistochemistry staining. R.T., N.M. and B.A. performed statistical analysis and interpreted the data. R.T., N.M. and B.A. wrote the paper. B.A. directed the overall project.

## Competing interests

N.P. is a consultant to the Empire Genomics. The remaining authors declare no competing interests.
