## [Peer Review File · Nature Communications]

Reviewers' Comments:

Reviewer #1:

Remarks to the Author:

The authors have to address the following point:

Introduction:

The authors should include a paragraph for SPINK1 description.

Figure 2.

The authors should show that the levels of SPINK1 decrease in 22RV1 and VCAP cells upon treatment with R1881. Does the concentration of R1881 influence SPINK1 expression?

g, h) The authors should verify the heat map data by QRT-PCR.

i-k) When the authors say: "...a remarkable increase both at transcript and protein levels was observed..." the authors have to present WB and MS data!

In general, the authors have to analyze the SPINK1 proteins levels in the different experiments by WB and MS. In the entire manuscript, the controls for immunofluorescence are missing! Please provide controls.

Figure 3.

b) The figure is not very informative. MACS called peaks should be shown in order to decipher from false positive peaks. The authors present 3 AREs around the SPINK1 gene (Fig. 3a) whereas several AR peaks appear in figure b. Please provide clarification. Please indicate in Fig. 3b where the AREs are located?

d-e) In addition, the authors have to carefully inspect and validate the state of transcription (initiation, elongation). To validate that H3K9ac levels decrease, the authors have to include/analyze the levels of Histone H3 as controls.

Supplementary 3d: Since the authors talk about a poised gene state they have to provide experimental data that this is the case. Please analyze the levels of histone marks corresponding to a poised state of genes.

f) Is repression dependent on the concentration of R1881 or DHT? Please present concentration dependent data!

Lane 271; page 11: the authors talk about H3K27ac: Where are the data shown?

Figure 4:

WB for SPINK1 knockdown are missing! All the data are based on mRNA expression levels.

It is not clear whether the authors performed RNA-seq. If yes, the authors should present a detailed analysis of the data. How many genes were differentially regulated, etc. Where does the EMT idea come from? This part of the manuscript is not well written and it is not self-explaining where the information is extracted from.

Supplementary Fig. 4c-e: The increase of doxorubicin, cisplatin and 5-fluorouracil efficacy in shSPINK-1 cells compared to control is not convincing.

c) The effect could also be due to an anti-proliferative effect of shSPINK-1. Thus, the authors should investigate the effect of shSPINK1 on 22RV1 proliferation. To confirm that stemness (self-renewal) is blocked the authors must isolate single cells from the primary assay and evaluate sphere-formation in a secondary assay.

Figure 5:

Is overexpression of SPINK1 in LNCaP or in any other prostate tumor cell line sufficient to induce a NEPC phenotype? Please provide this key data.

Reviewer #2:

Remarks to the Author:

Increasing number of prostate cancer patients who relapse after treatment with first or second generation of AR antagonists are found to show clinical features of neuroendocrine prostate cancer (NEPC). Understanding the molecular mechanism of castration resistant prostate cancer and NEPC

development possesses great value for diagnosis, prognosis and design of intervention approaches. Through public data analysis, biochemical research and examination on human patient samples, the authors in this paper revealed that AR and its co-repressor REST negatively regulated the expression of SPINK1. They further proposed that AR antagonists relieved the suppression of AR on SPINK1, and subsequently led to enhanced epithelial-mesenchymal-transition, drug-resistance, stemness, cellular-plasticity and neuroendocrine differentiation. In this paper, CHIP assays have been performed to demonstrate that AR and REST both bound to the SPINK1 promoter. In addition, a previously reported key factor of neuroendocrine differentiation, SOX2, was also suggested to contribute to the transcriptional regulation of SPINK1.

The paper is dealing with an interesting and clinically important scientific question. However, I found several major conclusions are not well-supported by the data presented. In addition, this manuscript lack sufficient mechanism exploration on how SPINK1 elicited downstream molecules in promoting cell plasticity and neuroendocrine differentiation.

There is no functional experimental evidence in vitro or in vivo to support that SPINK1 is indeed required for the cell plasticity, androgen antagonist resistance and neuroendocrine differentiation. Current data for the link between SPINK1 and NEPC is really weak. The only experiment that they have done for that purpose is a neurite length measurement assay upon SPRINK1 knockdown. As a matter of fact, neurite length is not really a reliable parameter to reflect neuroendocrine differentiation to begin with. And the SPINK1 knockdown efficiency in that experiment is not high enough.

Here are major concerns:

1. The protein levels in most of the experiments were not quantified or only shown by immunostaining. The authors need to provide immunoblotting data for SPINK1, E-CAD, VIMENTIN, SYP, SOX2 etc. to reach a solid conclusion.
2. 22RV1, as an AR-V7 positive cell line, is not proper to be used for AR related experiment to test response to R1881 and enzalutamide. Why did the author use 22RV1 in figure 1-4 for mechanistic study on AR and SPINK1, then LNCaP in figure 5-6 for REST and SPINK2? Without immunoblotting data, it is hard to say that LNCaP is "negative" of SPINK1, while 22RV1 is "positive".
3. It has been reported that paracrine of SPINK1 from prostate stromal cells can promote prostate cancer cell aggressiveness, particularly chemoresistance (Chen et al, Nat Commun, 2018, 9:4315). Authors should confirm whether elevated SPINK1 expression in cancer cells after ADT is caused by a relief of transcriptional repression of SPINK1 via AR in cancer cell itself or by an increased secretion of SPINK1 from stromal cells in vivo.
4. The function of SPINK1 in drug-resistance and cellular-plasticity are only investigated by gene down-regulation assays, overexpression experiments should be added to strengthen the importance of SPINK1 in these processes. It is necessary to confirm that SPINK1 upregulation after ADT can induce EMT and NE-like phenotypes in vivo by IF/IHC staining assay of associated markers such as E-cad, Vimentin, Tuj1, SYP and so on, in prostatic orthotopic or subcutaneous xenograft mouse model, or in prostatic spontaneous tumorigenic mouse model (eg. prostate specific PTEN null mice).
5. Given the protein encoded by SPINK1 is a secreted protein, the authors should further elucidate how SPINK1 acts on proliferation, migration and even lineage transition through autocrine or paracrine action modes.
6. In figure 1b, the human samples, especially the AR+ sample, did not look like prostate cancer but more like hyperplasia tissue.
7. In figure 2k, there seems to be a dose-dependent effect of Enza treatment on SPINK1 expression, although I cannot draw the same conclusion from the immunostaining pictures in fig2j. The RT-PCR result in Fig 2i did not show so either.
8. There is no information about the dose of drugs in Fig.2I
9. In Fig.3, the SPINK1 reporter luciferase assay was performed to demonstrate that AR negatively regulated the transcription of SPINK1. AR antagonists, in addition to R1881 should be used.
10. I am not sure why the authors suddenly switch to EMT in fig 4.

11. In the Fig.5, only SYP level was upregulated in the LNCap-AI, how about other markers such as NSE and CHGA. In addition, the impact of SPINK1 on NEPC marker expression (both in RNA and protein levels) was not examined.

12. The examination of SPINK1 in human NEPC patient or CRPC patient samples is needed to demonstrate that SPINK1 positively correlates with castration resistance and neuroendocrine differentiation.

13. iCK1 is an inhibitor for CK1, which can affect a number of other downstream effectors besides REST. REST knockdown or overexpression experiment should be performed to substantiate the conclusion that REST acted a co-suppressor of AR in repressing SPINK1 transcription.

Minor:

1) Authors should give a general introduction on the physiological function of SPINK1 and its related downstream pathways.

2) The endogenous expression of SPINK1 should be checked not only at mRNA level but also at protein level in prostate cancer cell lines to determine cell lines used for further investigation in vitro.

3) In Fig 4., the authors show that knockdown SPINK1 reverse epithelial-mesenchymal transition (EMT) and "a significant decrease in the number and size of the prostatospheres was observed in shSPINK1-1, shSPINK1-2 cells as compared to the control cells". However, it should be noted that mesenchymal-like prostate cancer cell subpopulation has actually been show to reduce self-renewal ability compared with E-cadherin higher sub-population (J Clin Invest, 2012). The authors should provide an explanation for their observation.

Reviewer #3:

Remarks to the Author:

The manuscript shows that androgen deprivation therapy may lead to the neuroendocrine differentiation of prostate cancer by upregulating the expression of SPINK1. The study is straightforward and the experiments are designed logically. It would be of considerable interest to the journal audience. A few concerns are noted below:

Major concerns:

1) Throughout the manuscript the authors used various methods other than Western blotting to assess the expression of SPINK1. It would be beneficial if the protein levels of SPINK1 are shown in experiments using shRNA against SPINK1 as well as overexpression of SPINK1.

2) The authors show that AR controls the expression of SPINK1 by binding to the promoter. Since SPINK1 is proposed to have an important role in castration resistance and therapy resistance, the effect of AR splice variants on the expression of SPINK1 should also be investigated.

3) The role of SPINK1 in prostate cancer cell proliferation or tumorigenesis should also be established to present functional evidence of the importance of SPINK1.

4) Given that SPINK1 expression promotes the expression of NE markers, the authors should present evidence of the levels of SPINK1 expression in samples of NE Prostate cancer.

5) The authors show that downregulation of SPINK1 confers resistance to chemotherapeutics such as doxorubicin. Since taxane agents such as docetaxel and cabazitaxel are used as first and second line chemotherapeutics against prostate cancer, the authors should examine whether SPINK1 plays a role in resistance to these agents.

6) The authors used 22Rv1 cell line for downregulation of SPINK1 expression. As this cell line has higher levels of expression of AR variants than that of the full length AR which might confound the results and to minimize cell line-specific effects, these experiments should also be repeated in another cell line.

7) The rationale for using LNCaP cells for generating shSPINK1 cells is not clear, given that the authors state that LNCaP cells are SPINK1 negative. Please explain.

8) The authors used LNCaP-derived ENZR cells generated by serial passaging as xenografts. These cells appear to show lower levels of AR activity and increased NE markers, compared to other ENZ-

resistant cells generated by other groups that exhibit higher levels of AR activity and no NE marker expression. This discrepancy should be addressed in the context of SPINK1 expression.

9) The authors state that downregulation of SOX2 using siRNA did not have any effect on the expression of SPINK1, whereas overexpression of SOX2 increases SPINK1 expression. This appears to be a contradictory result, if we are to assume that SOX2 controls SPINK1 expression. Please explain or otherwise address this discrepancy.

Minor concerns:

10) The discussion section should provide references for statements about SPINK1 expression in prostate cancers.

Authors responses to reviewers' comments

Reviewer #1: Expertise: AR signalling, molecular biology

Comment 1: Introduction: The authors should include a paragraph for SPINK1 description.

Response: We thank the reviewer for his/her efforts in thoroughly reviewing our manuscript and providing constructive suggestions. In the revised manuscript, we have now included the information regarding the role of SPINK1 in normal physiological condition as well as in cancers (*please see Introduction page number: 3*).

Comment 2: Figure 2: The authors should show that the levels of SPINK1 decrease in 22RV1 and VCaP cells upon treatment with R1881. Does the concentration of R1881 influence SPINK1 expression?

Response: In the older version of the manuscript, we have shown the concentration dependent effect of R1881 on *SPINK1* expression in 22RV1 cells by quantitative PCR. This data was provided in *Supplementary Figure 2b* in the older version of the manuscript. Following the reviewer's suggestion, we have now incorporated quantitative PCR data indicating the concentration dependent effect of R1881 on the *SPINK1* expression in VCaP cells (*Supplementary Figure 2e*). Similar to 22RV1 cells, we observed a concentration dependent decrease in the expression of *SPINK1* in R1881-stimulated VCaP cells. It was evident that 0.01 nM of synthetic androgen, R1881 was not effective in 22RV1 cells, while same concentration shows significant reduction in SPINK1 expression in VCaP cells, which are supposed to more responsive to androgen signaling.

Comment 3: Figure 2 (g, h) The authors should verify the heat map data by QRT-PCR.

Response: To address the reviewer's concern, we have now incorporated the quantitative PCR data for the genes shown in heatmap in *Figure 2g, h* in the older version of the manuscript. The quantitative PCR validated the expression of androgen-activated and androgen-repressed genes as indicated in the heatmaps wherein 22RV1 (GSE71797) and VCaP cells (GSE51872) were stimulated with R1881 and dihydrotestosterone (DHT), respectively. (*Please see Fig. 2g and Supplementary Figure 2f-i in the revised manuscript*).

Comment 4: Figure 2 (i-k) When the authors say: "...a remarkable increase both at transcript and protein levels was observed..."; the authors have to present WB and MS data! In general, the authors have to analyze the SPINK1 proteins levels in the different experiments by WB and MS. In the entire manuscript, the controls for immunofluorescence are missing! Please provide controls.

Response: We thank the reviewer for suggesting us to do Western blot. In the revised manuscript, we have now included Western blot data for the SPINK1 protein across different experiments. Prostate cancer cell line 22RV1 is known to have high endogenous levels of SPINK1^{1, 2, 3}. In the

revised manuscript, we have also provided the quantitative PCR and Western blotting data for the SPINK1 expression in a panel of prostate cancer cell lines (*Supplementary Fig. 2a, b*).

For the immunofluorescence controls, we have now included several cancer cell lines known to be SPINK1-positive, such as, WiDr (colorectal cancer)⁴, SK-MEL-173 (melanoma)³ and CAPAN-1 (pancreatic cancer)⁵, as positive controls for the SPINK1 expression (*Supplementary Fig. 2c*). Further, for negative controls, we have included immunofluorescence data for SPINK1 in PC3 cells (SPINK1-negative) and a no primary antibody control for SPINK1 staining in 22RV1 cells (*Supplementary Fig. 2c*).

To analyze the SPINK1 protein levels via mass spectrometry (MS), we performed the MS/MS based label-free quantification of 22RV1 cells (stimulated with R1881) and VCaP cells [treated with enzalutamide (Enza)], however, unfortunately, we did not observe any significant abundance of SPINK1 protein. Since SPINK1 is a small (~6.5kDa) secretory protein, it was challenging to detect its relative level via quantitative profiling of these cells. We also tried to perform SPINK1 relative quantitation via MS/MS, affinity-based enrichment, but unfortunately, this experiment was not successful due to lack of commercially available immunoprecipitation (IP) grade SPINK1 antibodies.

Finally, to support our findings, we have performed ELISA-based quantification of SPINK1 levels in the conditioned media (CM) and total cell lysate (CL) of the 22RV1 cells (stimulated with R1881) and VCaP cells (treated with Enza), using the human SPINK1-specific ELISA kit, which has been previously used by Chen et al, 2018⁶ (*Supplementary Fig. 2j, k*). Almost similar results were obtained by ELISA, as in quantitative PCR, immunostaining and Western blot.

Comment 5: Figure 3(b) The figure is not very informative. MACS called peaks should be shown in order to decipher from false positive peaks. The authors present 3 AREs around the SPINK1 gene (Fig. 3a) whereas several AR peaks appear in figure b. Please provide clarification. Please indicate in Fig. 3b where the AREs are located?

Response: We apologise to the reviewer for not providing the MACS called peaks in *Fig. 3b*. To identify the putative AR binding sites, we first scanned the *SPINK1* promoter for the presence of androgen response elements (AREs) by employing publicly available transcription factor binding prediction software, JASPAR (<http://www.jaspar.genereg.net>) and MatInspector (<http://www.genomatix.de>). Moreover, we also analyzed the Chromatin Immunoprecipitation-Sequencing (ChIP-Seq) data for the recruitment of AR on *SPINK1* promoter in androgen stimulated VCaP cells (GSE8428) (*Fig. 3b*). We identified 3 distinct binding sites for AR, namely, *ARE-1* and *ARE-2* which were predicted by JASPAR and MatInspector, whereas *ARE-3* was identified through ChIP-Seq analysis. As suggested by the reviewer, Model-based Analysis of ChIP-Seq data (MACS), a publicly available computational algorithm for peak calling in ChIP-Seq analysis which identifies the transcription factor binding sites across the genome was used⁷. In the revised manuscript, we are now showing the MACS called peaks for AR binding and location of 3 distinct AREs on the *SPINK1* promoter region (*Revised Fig. 3b*). We have also shown

MACS called peaks for the AR binding on *KLK3* and *NOV* promoters, which were used as controls in the ChIP assays.

Comment 6: Figure 3 (d-e) In addition, the authors have to carefully inspect and validate the state of transcription (initiation, elongation). To validate that H3K9ac levels decrease, the authors have to include/ analyze the levels of Histone H3 as controls.

Response: We agree with the reviewer's concern regarding the transcriptional state of the *SPINK1* upon androgen stimulation. It has been known that during transcription, the C-terminal domain (CTD) of the largest subunit of RNA polymerase II undergoes extensive post-transcriptional modifications. It comprises of multiple heptad repeats of YSPTSPS residues, and phosphorylation of these residues play a critical role in regulating different stages of transcription (e.g. initiation, elongation and termination)^{8,9}. The transcriptional initiation is marked by phosphorylation of the Ser5 residue of the CTD by CDK7, a component of Transcription Factor II H (TFIIH), and transcriptional elongation is marked by Ser2 phosphorylation by CDK9 (pTEFb)⁸.

Therefore as suggested by the reviewer, we investigated the transcriptional state of *SPINK1*, by performing chromatin immunoprecipitation-quantitative PCR (ChIP-qPCR) for the transcription initiation specific RNA polymerase II CTD Ser5 phosphorylation (p-Pol-II-S5) and transcription elongation specific RNA polymerase II CTD Ser2 phosphorylation (p-Pol-II-S2) for the *SPINK1* promoter region in androgen stimulated 22RV1 cells. Remarkably, along with the significant decrease in the recruitment of total RNA Pol II (Pol-II), a robust decrease in the enrichment of Ser5 and Ser2 phosphorylation of the RNA polymerase II CTD on the *SPINK1* promoter was noted, highlighting that both transcriptional initiation and elongation of the *SPINK1* gene was impaired (**Revised Fig. 3d**). Similarly, these marks were also examined for *KLK3* and *NOV* promoters, used as controls in this experiment (**Revised Supplementary Fig. 4b-d**).

Further, as suggested by the reviewer, in order to confirm the decrease in H3K9ac levels, we performed ChIP-qPCR for the total Histone H3 (H3) in androgen stimulated 22RV1 cells. As anticipated, there was no change in the total Histone H3 levels at the *SPINK1* promoter. (**Revised Supplementary Fig. 4e**). Likewise, no change in the total H3 protein levels by immunoblotting was observed in androgen stimulated 22RV1 cells (**Revised Supplementary Fig. 4e**).

Comment 7: Supplementary 3d: Since the authors talk about a poised gene state, they have to provide experimental data that this is the case. Please analyze the levels of histone marks corresponding to a poised state of genes.

Response: We are thankful to the reviewer for asking this critical piece of data, as this information further strengthen our finding. *The poised or bivalent promoters/enhancers are marked by the presence of both activating (H3K4me3) and repressive histone marks (H3K27me3)*^{10,11}. Briefly, H3K4me3 marks recruit various chromatin remodelers and histone acetylases and promote transcription, while H3K27me3 marks compact the chromatin and are involved in the negative regulation of the target genes¹⁰. Hence, to investigate whether *SPINK1* promoter is in a transcriptionally poised-state in VCaP cells, we performed ChIP-qPCR for the recruitment

H3K4me3 and H3K27me3 histone marks. Notably, simultaneous presence of both the marks (H3K4me3 and H3K27me3) on the *SPINK1* promoter in VCaP cells confirmed the poised state of this gene (**Supplementary Fig. 4g, h**). Intriguingly, we observed a significant increase in the repressive H3K27me3 marks in androgen stimulated 22RV1 cells, but no change in the activating H3K4me3 marks, thus confirming the AR signaling mediated transcriptional repression of *SPINK1* (**Supplementary Fig. 4g, h**). Similar pattern in the repressive/activation marks was also observed for androgen-repressed gene *NOV*. Conversely, *KLK3* being transcriptionally active, exhibits enrichment of H3K4me3 and no change in H3K27me3 marks (**Revised Supplementary Fig. 4g, h**).

Comment 8: Figure 3f) Is repression dependent on the concentration of R1881 or DHT? Please present concentration dependent data!

Response: We thank the reviewer for this constructive suggestion. We have now incorporated the data showing concentration dependent effect of R1881 on the luciferase reporter activity of the proximal and distal *SPINK1* promoter (SPINK1-PP and SPINK1-DP). As anticipated, a concentration dependent decrease in the promoter reporter activity of both SPINK1-PP and SPINK1-DP was observed in 22RV1 cells stimulated with 0.1, 1 and 10nM R1881 (**Fig. 3g**). PSA promoter was used as a positive control for testing a range of R1881 concentrations in this experiment (**Revised Supplementary Fig. 4i**).

Comment 9: Lane 271; page 11: the authors talk about H3K27ac: Where are the data shown?

Response: We apologise the reviewers for this misunderstanding. It was a typological error, it should be H3K9Ac.

Comment 10: Figure 4: WB for SPINK1 knockdown are missing! All the data are based on mRNA expression levels.

Response: We regret for not providing the WB data in Figure 4. In the revised manuscript, we have now added SPINK1 immunoblotting data showing knockdown of *SPINK1* in 22RV1-shSPINK1 cells as compared to 22RV1-shSCRM control cells (**Fig. 4a**).

Comment 11: It is not clear whether the authors performed RNA-seq. If yes, the authors should present a detailed analysis of the data. How many genes were differentially regulated, etc. Where does the EMT idea come from? This part of the manuscript is not well written, and it is not self-explaining where the information is extracted from.

Response: We apologise to the reviewer for the lack of clarity. To elucidate the biological processes governed by *SPINK1* in PCa, we have performed microarray based global gene expression profiling of stable 22RV1-shSCRM (control) and 22RV1-shSPINK1 cells (shSPINK1-1, shSPINK-2 and shSPINK-3). Our analysis revealed 697 genes downregulated in 22RV1-shSPINK1 cells (\log_2 fold change >0.5 or <-0.5 , 90% confidence interval), which were further used for pathway enrichment analysis ($P<0.05$) using DAVID (Database for Annotation,

Visualization and Integrated Discovery) (as shown in **Fig. 4b**). This information is now updated in the **Methods** section of the revised manuscript. The gene expression microarray data from this study has been submitted to the NCBI Gene Expression Omnibus (GEO, <http://www.ncbi.nlm.nih.gov/geo/>) under the accession number GSE124345.

The role of SPINK1 in eliciting EMT has been previously established by several independent groups^{2, 12, 13, 14}. Based on reviewer's comment, we have now re-written this section of the results to provide more clarity (page number 11 of the revised manuscript).

Comment 12: Supplementary Fig. 4c-e: The increase of doxorubicin, cisplatin and 5-fluorouracil efficacy in shSPINK-1 cells compared to control is not convincing.

Response: We thank the reviewer for expressing his/her concern. The **Supplementary Fig. 4c-e** is now labelled as **Supplementary Fig. 5d-f** in the revised manuscript. To emphasize on the significant difference between the IC₅₀ values of the doxorubicin, cisplatin and 5-fluorouracil in 22RV1-shSCRM *versus* 22RV1-shSPINK1 cells, we have included 95% confidence intervals (CI) of the IC₅₀ values in the revised manuscript (**Supplementary Fig. 5d-f**). The difference between the two groups is considered statistically significant, if the 95% CI of the IC₅₀ values do not intersect¹⁵. A significant increase in the chemosensitivity of shSPINK1 cells towards doxorubicin (IC₅₀ shSCRM cells = 276.6nM; **95% CI = 244.7 to 312.8** vs. IC₅₀ shSPINK1 cells = 101.6nM; **95% CI = 84.33 to 122.4**) and 5-fluorouracil (IC₅₀ shSCRM cells = 32.32 μM; **95% CI = 24.51 to 42.63 μM** vs. IC₅₀ shSPINK1 cells = 14.22 μM; **95% CI = 11.14 to 18.15 μM**) was noted as there was no intersection between the 95% CI of the IC₅₀ values in both 22RV1-shSPINK1-1 cells and 22RV1-shSCRM cells. Also, for cisplatin (IC₅₀ shSCRM cells = 16.72 μM; **95% CI = 12.32 to 22.70 μM** vs. IC₅₀ shSPINK1 cells = 9.8 μM; **95% CI = 7.491 to 12.84 μM**), although a slight overlap between 95% CI was observed, however, the difference between means of IC₅₀ values of 22RV1-shSCRM and 22RV-shSPINK1 was statistically significant (calculated using unpaired Student's *t* test). Hence our data show enhanced chemosensitivity in the 22RV1-shSPINK1-1 cells relative to control 22RV1-shSCRM cells.

Comment 13: Figure 4c) The effect could also be due to an anti-proliferative effect of shSPINK-1. Thus, the authors should investigate the effect of shSPINK1 on 22RV1 proliferation. To confirm that stemness (self-renewal) is blocked the authors must isolate single cells from the primary assay and evaluate sphere-formation in a secondary assay.

Response: We understand reviewer's concern regarding the role of SPINK1 in cell proliferation and stemness. Previously, we showed that stable *SPINK1*-silenced (shSPINK1) 22RV1 cells exhibit decrease in cell proliferation and number of colonies in anchorage-independent soft agar assay compared to the control shScrambled (shSCRM) cells².

In this study, we performed prostatosphere assay, a readout of stemness and self-renewal, which is a conventional method for assessing tumor sphere formation ability¹⁶. As mentioned in the **Methods** section of the manuscript, 22RV1-shSCRM and -shSPINK1 cells (1×10⁴) were plated in low adherence 6-well cell culture dishes in serum-free DMEM-F12 media (1:1) supplemented

with B27 (1X), EGF (20 ng/ml) and FGF (20 ng/ml). After 5-6 days, the prostatospheres formed were collected by gentle centrifugation, and mechanically dissociated into single cell suspension, followed by re-plating in fresh culture media. Next, the new prostatospheres formed, were again passaged in the similar manner for multiple generations (every 3rd day), and the experiment was terminated after two weeks. The prostatospheres formed were assessed for sphere forming efficiency. Mean area of spheres was measured using ImageJ software, and the spheres >50µm in diameter were counted and the values were represented as percent sphere formation efficiency.

We would like to mention that similar protocols have been used in numerous independent studies to evaluate the stemness potential of the genetically manipulated cancer cells^{3, 17, 18, 19}. Besides prostatosphere assay, we also performed assays for ALDH activity, Hoechst 33342 efflux-based side population (SP) and surface expression of stem cell marker CD117 (c-KIT) and the findings of these assays indicate the implication of SPINK1 in imparting stemness and self-renewal.

We hope these findings are sufficient to convince the reviewer that outcome of the prostatospheres assay is indeed not due to the anti-proliferative effect of shSPINK1.

Comment 14: Figure 5: Is overexpression of SPINK1 in LNCaP or in any other prostate tumor cell line sufficient to induce a NEPC phenotype? Please provide this key data.

Response: We thank the reviewer for suggesting this critical experiment. To investigate the possible association between SPINK1 and NE-like phenotype, we ectopically overexpressed SPINK1 in LNCaP cells, however, no change in the levels of NEPC markers was noted (data not shown). Nevertheless, SPINK1 overexpressing LNCaP cells show a marked increase in the migratory properties, ALDH activity and percentage of c-KIT-positive cell population (*revised Fig. 4h-j*), confirming the predominant role of SPINK1 in EMT, stemness and cellular plasticity. Intriguingly, in another set of experiment, wherein *SPINK1* was silenced using shRNA-mediated approach in LNCaP cells, referred as LNCaP-shSPINK1 cells, when subjected to NE-transdifferentiation in androgen-deprived conditions (*revised Fig. 5j*) resulted in significant reduction in the length of neurite-like projections, accompanied with downregulation of classical EMT (E-Cad, Vimentin and N-Cad) as well as NEPC (SYP and ENO2) markers (*revised Fig. 5k-m*). Importantly, we have also provided additional data on NEPC patients, wherein elevated SPINK1 levels in a subset of NEPC clinical specimens, further confirms its undeniable association with NE-like features in prostate cancer (*revised Fig. 7e and Supplementary Fig. 9c*).

Reviewer #2: Expertise: Prostate cancer molecular biology, resistance

Increasing number of prostate cancer patients who relapse after treatment with first or second generation of AR antagonists are found to show clinical features of neuroendocrine prostate cancer (NEPC). Understanding the molecular mechanism of castration resistant prostate cancer and NEPC development possesses great value for diagnosis, prognosis and design of intervention approaches. Through public data analysis, biochemical research and examination on human patient samples, the authors in this paper revealed that AR and its co-repressor REST negatively regulated the expression of SPINK1. They further proposed that AR antagonists relieved the suppression of AR on SPINK, and subsequently led to enhanced epithelial-mesenchymal-transition, drug-resistance, stemness, cellular-plasticity and neuroendocrine differentiation. In this paper, CHIP assays have been performed to demonstrate that AR and REST both bound to the SPINK1 promoter. In addition, a previously reported key factor of neuroendocrine differentiation, SOX2, was also suggested to contribute to the transcriptional regulation of SPINK1.

The paper is dealing with an interesting and clinically important scientific question. However, I found **several major conclusions are not well-supported by the data presented**. In addition, this manuscript lack sufficient mechanism exploration on how SPINK1 elicited downstream molecules in promoting cell plasticity and neuroendocrine differentiation.

Response: We thank the reviewer for thoroughly going through the manuscript and providing constructive suggestions, which were instrumental in improving the quality of the manuscript. As mentioned by the reviewer that “several major conclusions are not well-supported by the data presented” now in the revised manuscript we have added considerable amount of crucial data which further strengthen the SPINK1-mediated cellular plasticity and neuroendocrine differentiation.

There is no functional experimental evidence in vitro or in vivo to support that SPINK1 is indeed required for the cell plasticity, androgen antagonist resistance and neuroendocrine differentiation. **Current data for the link between SPINK1 and NEPC is really weak**. The only experiment that they have done for that purpose is a neurite length measurement assay upon SPINK1 knockdown. As a matter of fact, neurite length is not really a reliable parameter to reflect neuroendocrine differentiation to begin with. And the SPINK1 knockdown efficiency in that experiment is not high enough.

Response: We are thankful to the reviewer for suggesting new experiments, which provided convincing evidences to prove an association between SPINK1 and NEPC. In the revised manuscript, we have also shown upregulation of SPINK1 and NEPC markers in the orthotopic VCaP xenografts and subcutaneous 22RV1 xenografts excised from the AR-antagonists treated orchiectomized mice.

Here are major concerns:

Comment 1: The protein levels in most of the experiments were not quantified or only shown by immunostaining. The authors need to provide immunoblotting data for SPINK1, E-CAD, VIMENTIN, SYP, SOX2 etc. to reach a solid conclusion.

Response: We apologize the reviewer for not providing the immunoblotting data. As suggested, we have now performed immunoblot experiments for the markers pointed by the reviewer. For SPINK1, we have used monoclonal antibody previously used by Yu Chen's group²⁰. Western blot data has been incorporated in the revised manuscript as follows:

- 1) Immunoblots for SPINK1 levels for all the experiments throughout the manuscript.
- 2) Immunoblots for E-Cad and Vimentin in 22RV1-shSCRM and 22RV1-shSPINK1 cells and LNCaP-AI shSCRM and LNCaP-AI-shSPINK1 cells (*Fig 4a and Fig. 5l*).
- 3) Immunoblots for SYP levels in LNCaP-AI and LNCaP-AI-shSCRM and LNCaP-AI-shSPINK1 cells (*Fig. 5d, m*).
- 4) Immunoblots for SOX2 levels in LNCaP-AI cells, siRNA-mediated SOX2-silenced LNCaP, 22RV1 cells and, LNCaP cells with transient overexpression of SOX2 (*Fig. 5d and Fig. 6e-g*).

Comment 2: 22RV1, as an AR-V7 positive cell line, is not proper to be used for AR related experiment to test response to R1881 and enzalutamide. Why did the author use 22RV1 in figure 1-4 for mechanistic study on AR and SPINK1, then LNCaP in figure 5-6 for REST and SPINK1? Without immunoblotting data, it is hard to say that LNCaP is “negative” of SPINK1, while 22RV1 is “positive”.

Response: We understand reviewer's concern and apologise for the inconvenience. In the revised manuscript, we have now included the immunoblot data for SPINK1 in a panel of prostate cancer cell lines, as indicated previously, 22RV1 cells show highest endogenous expression of SPINK1 followed by VCaP cells with moderate levels, while LNCaP cells were negative for SPINK1 (*revised Supplementary Fig. 2a-b*).

Previously, many independent studies have used 22RV1 cells for androgen stimulation experiments^{21, 22}. In this manuscript, we have used 22RV1 for androgen stimulation experiment, because of its SPINK1-positive nature. As the reviewer could appreciate data shown in *Fig. 2a-c*, wherein upon androgen stimulation SPINK1 levels were gone down in 22RV1 cells, and conversely, *KLK3*, a well-known androgen responsive gene shows upregulation. We have validated these findings in VCaP cells (low SPINK1 levels), which upon anti-androgen treatment show a significant increase in SPINK1 expression (*Fig. 2h-j and Supplementary Fig. 2k*).

We would also like to draw attention of the esteemed reviewer's to the heatmap shown in *Fig. 2g*, where gene expression profiling of the 22RV1 cells (GSE71797) and VCaP (GSE51872) show similar pattern of the genes up- and down-regulated upon androgen stimulation.

To understand the AR-REST mediated regulation of *SPINK1*, we chose LNCaP cells, which has high endogenous expression of REST²³, and is negative for SPINK1 expression

(*Supplementary Fig. 2a-b*). Also, LNCaP cells being highly responsive to androgen-signaling, is a preferred cell line for modelling NE-transdifferentiation²⁴. To study the role of SPINK1 in neuroendocrine transdifferentiation, we cultured LNCaP cells in androgen-deprived media for 30 days and generated LNCaP-AI cells, which show a remarkable decrease in the REST expression accompanied with robust increase in SPINK1 levels (*Fig. 5c, d*). Further, to support our findings, we performed stable knockdown of *REST* in LNCaP cells, and a significant increase in the expression of SPINK1 as compared to the control cells was observed (*Supplementary Fig. 8k*), thus confirming the AR-REST mediated transcriptional repression of *SPINK1*.

Additionally, to rule out the possible role of AR-V7 in the regulation of SPINK1, we have also generated dox-inducible LNCaP AR-V7 cells, and interestingly dox-inducible LNCaP AR-V7 due to consecutive AR-signaling failed to show upregulation of SPINK1 upon 40-days androgen deprivation experiment (*revised Fig. 5f-i*).

Comment 3: It has been reported that paracrine of SPINK1 from prostate stromal cells can promote prostate cancer cell aggressiveness, particularly chemoresistance (Chen et al, Nat Commun, 2018, 9:4315). Authors should confirm whether elevated SPINK1 expression in cancer cells after ADT is caused by a relief of transcriptional repression of SPINK1 via AR in cancer cell itself or by an increased secretion of SPINK1 from stromal cells in vivo.

Response: We sincerely appreciate the reviewer's concern in context of elevated SPINK1 expression, whether it is caused by relieved transcriptional repression of *SPINK1* via AR in cancer cells itself or by an increased secretion of SPINK1 from the stromal cells. The reviewer correctly highlighted the paracrine role of SPINK1 in promoting cancer cell aggressiveness, which has been recently demonstrated by Chen et al⁶. In this study, the authors have shown the *increased SPINK1 levels in the stromal cells and serum of prostate cancer patients*, who underwent chemotherapy compared to untreated patients. However, in our study we have observed the inverse association of SPINK1 and AR expression in the cancer cells of prostate cancer patients (*Fig. 1a-d*). We have also shown elevated SPINK1 levels in the cancer cells of AR antagonists treated mice xenografts (*Fig. 7a-e*). Furthermore, in the revised manuscript we have also shown elevated levels of SPINK1 in the PCa specimens of NEPC patients (*Fig. 7f*). Taken together, our findings highlight that androgen-deprivation mediated upregulation of SPINK1 expression in the prostate cancer cells.

Comment 4. The function of SPINK1 in drug-resistance and cellular-plasticity are only investigated by gene down-regulation assays, overexpression experiments should be added to strengthen the importance of SPINK1 in these processes.

Response: We thank the reviewer for this suggestion. In the revised manuscript, we have now incorporated this data, wherein overexpression of SPINK1 in LNCaP cells leads to robust increase in the migratory properties (*Revised Fig. 4h and Supplementary Fig. 5g*). Additionally, overexpression of SPINK1 in these cells also results in a significant increase in stemness, as depicted by an increased aldehyde dehydrogenase (ALDH) activity and CD117 (c-KIT) positive

cell population (*Revised Fig. 4i, j*). These evidences further confirm the role of SPINK1 in EMT, stemness and cellular plasticity.

Comment 5: It is necessary to confirm that SPINK1 upregulation after ADT can induce EMT and NE-like phenotypes in vivo by IF/IHC staining assay of associated markers such as E-cad, Vimentin, Tuj1, SYP and so on, in prostatic orthotopic or subcutaneous xenograft mouse model, or in prostatic spontaneous tumorigenic mouse model (e.g. prostate specific PTEN null mice).

Response: We highly appreciate reviewer's remark to confirm the SPINK1 upregulation in response to ADT using mice xenograft model. Following reviewer's advice, VCaP tumors generated by orthotopic implantation of cells in immunodeficient (HSD/athymic nude– *Foxn1^{nu}*) mice, and were administered with either vehicle or AR-antagonists [enzalutamide or apalutamide (ARN-509)]²⁵. These tumors were characterized by RNA-Seq and immunostaining for the SPINK1 expression, NEPC and EMT markers. Interestingly, we observed a significant increase in the expression SPINK1 along with NEPC [SYP, CHGA and TUBB3 (Tuj1)] and mesenchymal (VIM) markers (*Fig. 7b, c*). Intriguingly, an increase in the E-Cad (*CDH1*) expression was also observed upon treatment with AR-antagonists (*Supplementary Fig. 9a, b*), which is in agreement with a recent contradictory report wherein E-Cad is shown to functions as a survival-factor and supports metastases in mice model²⁶.

Additionally, we also developed 22RV1 xenograft model in immunodeficient intact or castrated mice (CrI:CD1-*Foxn1^{nu}*), wherein administration of enzalutamide led to a significant increase in the expression of SPINK1 along with other NEPC markers (SYP and TUBB3) (*Fig. 7d, e*). Conclusively, our mice xenograft data show elevated SPINK1 levels in AR-antagonists treated mice, further strengthening our claim that ADT-induced SPINK1 upregulation is associated with NE-phenotype.

Comment 6: Given the protein encoded by SPINK1 is a secreted protein, the authors should further elucidate how SPINK1 acts on proliferation, migration and even lineage transition through autocrine or paracrine action modes.

Response: The role of SPINK1 in eliciting proliferation, migration and autocrine-paracrine signaling has been previously elucidated by several independent groups^{2, 3, 4, 12, 13}. We have now included this critical information in the *Introduction* section of the revised manuscript.

Comment 7: In Figure 1b, the human samples, especially the AR+ sample, did not look like prostate cancer but more like hyperplasia tissue.

Response: The *Fig. 1b* represents IHC staining for the SPINK1 and AR in SPINK1-positive and SPINK1-negative prostate cancer patients. All pathological grading of the TMAs has been performed by the board-certified pathologist, and that particular patient's specimen corresponds to Gleason score 7. However, as pointed out by the esteemed reviewer, we have now replaced the PCa-1 image for AR IHC staining in the revised *Fig. 1b and Supplementary Fig 1a*.

Comment 8: In figure 2k, there seems to be a dose-dependent effect of Enza treatment on SPINK1 expression, although I cannot draw the same conclusion from the immunostaining pictures in Fig 2j. The RT-PCR result in Fig 2i did not show so either.

Response: We thank the reviewer for pointing this out. In revised manuscript, *Fig. 2h-j* represents the expression of SPINK1 in enzalutamide treated VCaP cells at the indicated concentrations. However, as depicted from the quantitative PCR, immunoblotting and immunofluorescence experiments, concentrations 2.5 μ M and 5 μ M of enzalutamide were equally efficacious in elevating the SPINK1 levels in VCaP cells (*Revised Fig. 2h-j*).

Comment 9: There is no information about the dose of drugs in Fig. 2l.

Response: The *Fig. 2l* represents *SPINK1* and *KLK3* expression in 22RV1 cells initially primed with either with 10nM of R1881 or 10 μ M of Enza for 3 days, followed by Enza treatment or R1881 stimulation for the next 3 days. The information regarding the concentrations of R1881 and enzalutamide is mentioned in the figure legends, nonetheless, we have also included this information in the revised main manuscript (*Page number: 8*).

Comment 10: In Fig.3, the SPINK1 reporter luciferase assay was performed to demonstrate that AR negatively regulated the transcription of SPINK1. AR antagonists, in addition to R1881 should be used.

Response: We thank the reviewer for this valuable suggestion. In the revised manuscript, we have now incorporated the data showing the effect of enzalutamide treatment on the promoter reporter activity of both SPINK1-PP and SPINK1-DP luciferase constructs. Interestingly, we observed a significant decrease in the luciferase activity of both the reporter constructs upon enzalutamide treatment with respect to control (*Revised Fig. 3h*). We have used PSA or *KLK3* promoter as a positive control for this experiment (*Revised Supplementary Fig. 4j*)

Comment 11: I am not sure why the authors suddenly switch to EMT in fig 4.

Response: In *Fig. 4*, we have shown the downstream effects of SPINK1 in eliciting various oncogenic traits such as, EMT, drug-resistance and stemness, thus, confirming the role of SPINK1 in tumorigenesis. In the revised manuscript, now we have provided the rationale of investigating SPINK1-mediated EMT and stemness (*under Fig. 4 results on Page number: 11*).

Comment 12: In the Fig.5, only SYP level was upregulated in the LNCaP-AI, how about other markers such as NSE and CHGA. In addition, the impact of SPINK1 on NEPC marker expression (both in RNA and protein levels) was not examined.

Response: We thank the reviewer for expressing his/her concern about other NEPC markers. As desired, in the revised manuscript, we have now incorporated the immunoblot data for NSE or ENO2 expression in LNCaP-AI cells (*Fig. 5d*). Unfortunately, the CHGA antibody did not work in the immunoblot experiments (Santa Cruz, sc-47714). However, we have examined the

expression of CHGA in the VCaP and 22RV1 mice xenografts by using IHC grade CHGA antibody (**Revised Fig. 7b, d**).

To understand the effect of SPINK1 on the expression of NEPC markers, we established stable LNCaP cell line transduced with lentiviral particles containing SPINK1-shRNA (LNCaP-shSPINK1) and cultured them in androgen-deprived condition for 30 days (**Revised Fig. 5j**). Our aim for conducting this experiment was to silence *SPINK1* expression in LNCaP-AI cells which upon NE-transdifferentiation show increased SPINK1 expression.

Intriguingly, androgen-deprived LNCaP-AI-shSPINK1 cells show reduced expression of SPINK1, NEPC (*SYP* and *ENO2*) and EMT (*E-Cad*, *Vimentin* and *N-Cad*) markers as compared to LNCaP-AI-shSCRM cells (**Revised Fig. 5l, m**). Further, siRNA-mediated knockdown of *SPINK1* in 22RV1 cells led to a reduced expression of *SYP* (**Revised Supplementary Fig. 5h**) and reduced surface expression of *NCAM1* (**Revised Fig. 4k**). Additionally, we also performed siRNA-mediated knockdown of *SPINK1* in 42D^{ENZR} and 42F^{ENZR} cells, derivatives of LNCaP-derived CRPC cell lines, established via multiple serial transplantation of the enzalutamide-resistant tumors in male athymic mice²⁷. Interestingly, a significant decrease in the expression of *SYP* in both cell lines, while reduced CHGA level in 42F^{ENZR} cells was noted (**Supplementary Fig. 7g, h**). Thus, our new data included in the revised manuscript highlight the impact of SPINK1 on the expression of NE markers.

Comment 12: The examination of SPINK1 in human NEPC patient or CRPC patient samples is needed to demonstrate that SPINK1 positively correlates with castration resistance and neuroendocrine differentiation.

Response: We thank the reviewer for this valuable suggestion. Previous studies have already shown that elevated levels of SPINK1 have been associated with shorter time to castrate resistant prostate cancer (CRPC)^{20, 28, 29}. Moreover, a recent report indicated ~7% of the neuroendocrine PCa patients were positive for SPINK1 expression³⁰. As suggested by the reviewer, we examined the expression of *SPINK1* in the RNA-Seq data of the Beltran cohort³¹, and 8 out of 36 NEPC patients showed increased expression of *SPINK1* (**Revised Supplementary Fig. 9c**). Next, to validate the expression of SPINK1, AR and NE-markers in these NEPC patients, we have chosen NEPC cases on the basis of SPINK1-high and -low status, namely, WCM12, who underwent radical prostatectomy and responded well to platinum-based therapy³²; WCM155, who developed lung and liver metastases after 12 months of primary ADT and responded well to alisertib³³, and WCM677, treated with ADT and radium, and harbored somatic alterations in *RBI*, *PTEN* and *BRCA2*³¹. Notably, similar to our SPINK1 and AR IHC data in prostate adenocarcinoma (**Fig. 1**), WCM12 also show positive staining for SPINK1 and negative for AR expression. While, WCM155, a patient-derived organoid exhibit weak cytoplasmic staining for SPINK1 and negative for AR expression. Conversely, WCM677 showed negative staining for SPINK1 expression and focal weak positive staining for AR (**Fig. 7f**). These findings thus, highlights the importance of SPINK1 in ADT or therapy induced NEPC progression.

Comment 13: iCK1 is an inhibitor for CK1, which can affect a number of other downstream effectors besides REST. REST knockdown or overexpression experiment should be performed to substantiate the conclusion that REST acted a co-suppressor of AR in repressing SPINK1 transcription.

Response: We understand the reviewer's concern regarding the possible role of iCK1 in affecting other downstream effectors besides REST. In *Fig. 6k* of the revised manuscript, Casein Kinase-1 inhibitor (iCK1) treated 22RV1 cells show reduced expression of SPINK1. To validate whether iCK1 mediated restoration of REST leads to *SPINK1* downregulation, we also checked for the expression of other REST-target gene²³. Interestingly, a significant decrease in the expression of other REST-target genes namely, *SYP*, *BDNF*, *SYN1* and *GRIN2A* was observed in iCK1 treated 22RV1 cells (*Revised Fig. 6k and Supplementary Fig. 8i*).

As suggested by Reviewer, we performed ectopic overexpression of REST in 22RV1 cells, wherein a significant decrease in the SPINK1 expression was observed (*Supplementary Fig. 8j*). Conversely, *REST*-silenced LNCaP cells showed an increase in the expression of SPINK1 as well as other REST target genes (*Supplementary Fig. 8k, l*), which confirms the role of REST as a co-suppressor of AR in repressing *SPINK1* transcription.

Minor:

Comment 14: Authors should give a general introduction on the physiological function of SPINK1 and its related downstream pathways.

Response: We thank the reviewer for this suggestion. We have now included this information in the *Introduction* section of the revised manuscript.

Comment 15: The endogenous expression of SPINK1 should be checked not only at mRNA level but also at protein level in prostate cancer cell lines to determine cell lines used for further investigation in vitro.

Response: As suggested, we have now incorporated the immunoblotting data for SPINK1 in all the experiments throughout the revised manuscript.

Comment 16: In Fig 4., the authors show that knockdown SPINK1 reverse epithelial-mesenchymal transition (EMT) and "a significant decrease in the number and size of the prostatospheres was observed in shSPINK1-1, shSPINK1-2 cells as compared to the control cells". However, it should be noted that mesenchymal-like prostate cancer cell subpopulation has actually been showed to reduce self-renewal ability compared with E-cadherin higher sub-population (J Clin Invest, 2012). The authors should provide an explanation for their observation.

Response: We understand the reviewer's concern regarding the role of E-Cadherin (E-Cad) in self-renewal. In the above-mentioned study, Terrassa et al. (*J. Clin. Investigation, 2012*), demonstrated that a subpopulation cells with epithelial phenotype exhibits high expression of E-Cad, and shows enhanced stemness and self-renewal ability³⁴. However, E-Cad, an intracellular

adhesion protein, has been implicated majorly as a tumor-suppressor across multiple cancer types, and its loss is directly involved in promoting various oncogenic traits such as increase in proliferation, invasion, stemness and metastases^{35, 36, 37, 38}.

Similarly, our pathway enrichment analysis revealed ‘stem cell population maintenance’ as one of the pathways significantly downregulated in *SPINK1*-silenced 22RV1 cells (**Revised Fig. 4b**). Further, loss of *SPINK1* led to an increase in the expression of E-Cad (an epithelial marker) with concomitant loss of Vimentin (a mesenchymal marker) (**Revised Fig. 4a and Supplementary Fig. 5c**); and reduced stemness as shown by a significant decrease side-population, ALDH⁻ and c-KIT positive cells and a significant decrease in the number and size of the prostatospheres (**Revised Fig. 4c-e**). Although, all this data is generated using in-vitro cell-based assay. Intriguingly, a recent paper has also shown the tumor promoting role of E-Cad in invasive ductal carcinomas of breast, wherein it aids tumor growth and metastases²⁶. In concordance to this report, we also observed an enhanced expression of E-Cad in our VCaP xenografts excised from mice treated with AR-antagonists (**Supplementary Fig. 9a-b**).

Reviewer #3: Expertise: Prostate cancer molecular biology, resistance

The manuscript shows that androgen deprivation therapy may lead to the neuroendocrine differentiation of prostate cancer by upregulating the expression of SPINK1. The study is straightforward, and the experiments are designed logically. It would be of considerable interest to the journal audience. A few concerns are noted below:

Response: We thank the reviewer for his/her efforts in reviewing our manuscript, overall positive assessment and providing constructive suggestions.

Major concerns:

Comment 1: Throughout the manuscript the authors used various methods other than Western blotting to assess the expression of SPINK1. It would be beneficial if the protein levels of SPINK1 are shown in experiments using shRNA against SPINK1 as well as overexpression of SPINK1.

Response: As suggested by the reviewer, we have now incorporated the immunoblotting data for all the experiments for SPINK1 expression as well as for other relevant markers. However, ectopic SPINK1 overexpression in LNCaP cells was not detected by immunoblotting. We think that one of the possible reasons that monoclonal antibody for SPINK1 (R&D Systems, MAB7496-SP) failed to detect SPINK1 tagged with 2xV5 (construct used for overexpression of SPINK1), could be due to modification or disturbance in the SPINK1 epitope, thereby hindering the antibody binding with SPINK1 protein. Please note that the immunogen used for generating SPINK1 antibody (R&D Systems, MAB7496-SP) is an *E. coli* derived recombinant human SPINK1 (Asp24 – Cys79 (56 amino acids with accession number: P00995), which lies within the Kazal domain (Leu26–Cys79) of the SPINK1.

Comment 2: The authors show that AR controls the expression of SPINK1 by binding to the promoter. Since SPINK1 is proposed to have an important role in castration resistance and therapy resistance, the effect of AR splice variants on the expression of SPINK1 should also be investigated.

Response: We understand reviewer's concern that whether AR splice variants have any effect on *SPINK1* expression. To address this, we first examined the publicly available RNA-Seq dataset (GSE80743) where the authors have performed siRNA mediated knockdown of AR splice variants (*AR-V1*, *AR-V3*, *AR-V4* and *AR-V7*) in 22RV1 cells³⁹. Interestingly, we found a significant increase in the expression of *SPINK1* upon knockdown of AR splice variants as compared to the control (siNT) (**Revised Supplementary Fig. 3j**).

Since, long-term androgen deprivation in LNCaP cells results in increased SPINK1 levels in androgen-deprived LNCaP-AI cells (**Revised Fig. 5c-e**), we generated dox-inducible LNCaP AR-V7 overexpressing cells and subjected them to long-term androgen deprivation with or without doxycycline (dox), induced at day 10 and maintained dox treatment till 40th day (**Revised Fig. 5f**). Interestingly, dox induction led to a significant upregulation of AR-V7 expression in LNCaP cells,

leading to constitutively active androgen signaling. No increase in the SPINK1 expression was observed in dox-induced LNCaP AR-V7 cells, while uninduced control cells with no AR-V7 expression, show higher SPINK1 levels (**Revised Fig. 5g, h**). These findings highlight the fact that activation of AR signaling negatively regulates the SPINK1 expression and androgen ablation leads to SPINK1 upregulation.

Comment 3: The role of SPINK1 in prostate cancer cell proliferation or tumorigenesis should also be established to present functional evidence of the importance of SPINK1.

Response: We appreciate the reviewer's concern regarding the established role of SPINK1 in tumorigenesis. In the revised manuscript, we have included the normal physiological and tumor promoting role of SPINK1 in the *Introduction* section (**Page number: 3**). Previously, the role of SPINK1 as a pro-proliferative, pro-invasive and anti-apoptotic factor has been established by our group as well as by others^{2, 3, 4, 6, 13, 40}. We have also demonstrated the autocrine/paracrine function of SPINK1 in promoting tumor progression, and provided evidence that SPINK1-mediated oncogenicity is partly mediated through EGFR signaling².

Comment 4: Given that SPINK1 expression promotes the expression of NE markers, the authors should present evidence of the levels of SPINK1 expression in samples of NE Prostate cancer.

Response: We thank the reviewer for this important suggestion. We analyzed the RNA-seq data of the Beltran cohort³¹ for the *SPINK1* expression, and observed increased *SPINK1* expression in 8 out of 36 NEPC patients (**Revised Supplementary Fig. 9c**). We next classified these patients based on SPINK1-high and -low status, and selected small-cell NEPC cases namely, WCM12 and WCM155 being SPINK1-high, and WCM677 as SPINK1-low, and performed IHC for the expression of SPINK1, AR and NE markers. Interestingly, WCM12 was found to be SPINK1 positive and AR negative; WCM155, a patient-derived organoid showed weak cytoplasmic staining for SPINK1 and negative for AR, and WCM677 exhibited negative staining for SPINK1 and focal weak positive staining for AR (**Fig. 7f**). Taken together, these results show elevated levels of SPINK1 in a subset of NEPC cases, however, these findings need to be interrogated using larger NEPC cohort.

Comment 5: The authors show that downregulation of SPINK1 confers resistance to chemotherapeutics such as doxorubicin. Since taxane agents such as docetaxel and cabazitaxel are used as first- and second-line chemotherapeutics against prostate cancer, the authors should examine whether SPINK1 plays a role in resistance to these agents.

Response: We understand the reviewer's concern regarding the selection of doxorubicin, 5-fluorouracil (5-FU) and cisplatin, and not docetaxel in the drug sensitivity assay. Previously, we have shown that SPINK1 plays a key role in imparting chemoresistance in colorectal cancer, using anticancer drug doxorubicin⁴. To recapitulate this finding in prostate cancer background, we used doxorubicin, 5-FU and cisplatin for the chemosensitivity assay using 22RV1-shSCRM and 22RV1-shSPINK1 cells. These three anticancer drugs target the DNA through different

mechanisms, which further prompted us to use them for understanding the role of SPINK1 in mediating chemoresistance in PCa cells. Doxorubicin is a DNA intercalating agent which inhibits the progression of topoisomerase II, thus, halting transcription and cell replication. 5-FU is an antimetabolite drug, which gets incorporated into DNA and RNA and inhibits the essential biosynthetic pathways. Moreover, cisplatin is a platinum based anticancer agent, which crosslinks the DNA bases, interferes with DNA repair, thus, involved in DNA damage and apoptosis. Therefore, for the proof of the principle and to confirm our previous findings, we used these anticancer drugs for the chemosensitivity assay (*Supplementary Fig. 5d-f*). As mentioned by the reviewer that docetaxel is a taxane-based microtubule inhibitor, which leads to cell cycle arrest followed by apoptosis. We performed the drug sensitivity assay with docetaxel in 22RV1-shSCRM and -shSPINK1 cells, however, to our surprise no significant change in the IC₅₀ values of 22RV1-shSPINK1 cells compared to the control 22RV1-shSCRM cells was observed.

Comment 6: The authors used 22Rv1 cell line for downregulation of SPINK1 expression. As this cell line has higher levels of expression of AR variants than that of the full-length AR which might confound the results and to minimize cell line-specific effects, these experiments should also be repeated in another cell line.

Response: We performed downregulation of SPINK1 in 22RV1 cells as they show highest endogenous expression of SPINK1 (*Supplementary Fig. 2a-b*). However, to address reviewer's concern, we also established LNCaP-shSPINK1 cells, and subjected them to long-term androgen deprivation. As we know that androgen-deprived LNCaP cells (for 30 days) show a robust increase in SPINK1 expression (*Fig. 5c-e*), however stable LNCaP-shSPINK1 cells didn't show increase in SPINK1 levels during NE-transdifferentiation (due to shRNA-mediated knockdown of SPINK1). And LNCaP-shSPINK1 cells, which exhibit reduced SPINK1 levels as compared to LNCaP-shSCRM cells, show reduced neurite lengths, EMT and NEPC markers during NE-transdifferentiation (*Fig. 5k-m*), thus confirming its critical role in cellular plasticity and NEPC.

Comment 7: The rationale for using LNCaP cells for generating shSPINK1 cells is not clear, given that the authors state that LNCaP cells are SPINK1 negative. Please explain.

Response: LNCaP is an androgen responsive cell line which undergoes neuroendocrine (NE) transdifferentiation upon long-term androgen deprivation²⁴. Our data suggests that LNCaP cells do not express SPINK1 under normal condition (*Revised Supplementary Fig. 2a-b*), but upon NE-transdifferentiation, shows robust increase in SPINK1 expression (*Revised Fig. 5c-e*). Further, to understand the role of SPINK1 in mediating cellular plasticity and NE-transdifferentiation, we generated stable LNCaP-shSCRM and LNCaP-shSPINK1 cells and, subjected them to long-term androgen deprivation for 30 days (*Revised Fig. 5j*). Interestingly, the LNCaP-shSPINK1 cells expressing shRNA against *SPINK1* transcript, showed reduced expression of SPINK1 as compared to the LNCaP-shSCRM upon long-term androgen deprivation, and a significant reduction in the length of neurite-like projections, EMT and NEPC markers as compared to control LNCaP-

shSCRM cells (**Revised Fig. 5k-m**). These findings indicate the significance of SPINK1 in cellular plasticity and NEPC transdifferentiation.

Figure 1: Schematic showing stable LNCaP-shSCRM and LNCaP-shSPINK1 cells subjected to long term androgen deprivation (AD 30 days).

Comment 8: The authors used LNCaP-derived ENZR cells generated by serial passaging as xenografts. These cells appear to show lower levels of AR activity and increased NE markers, compared to other ENZ-resistant cells generated by other groups that exhibit higher levels of AR activity and no NE marker expression. This discrepancy should be addressed in the context of SPINK1 expression.

Response: We have used LNCaP-derived ENZ-resistant (ENZr) cells generated by Dr. Amina Zoubeidi’s group²⁷. During serial propagation of these tumors in mice, they obtained two types of ENZR xenografts, namely, PSA-positive (PSA⁺) xenografts with high AR signaling and PSA-negative (PSA⁻) xenografts with reduced AR signaling. The cell lines 42D^{ENZr} and 42F^{ENZr} are derivative of the PSA⁻ ENZR xenografts, whereas, cell lines 49C^{ENZr} and 49F^{ENZr} were derived from PSA⁺ ENZR xenografts. To validate the association of SPINK1 with NE-phenotype and AR independence, we have selected the 42D^{ENZr} and 42F^{ENZr} as these two cell lines exhibit high *SPINK1* expression, and more NE-like phenotype with reduced AR signaling (**Supplementary Fig. 7a-f**). Thus, based on these features, we selected 42D^{ENZr} and 42F^{ENZr}, which we believe would serve as a better model to understand the role of SPINK1-mediated regulation of cellular plasticity and NE-differentiation.

Figure 2: Bar plot showing relative expression of *SPINK1* in 42D^{ENZr} (PSA⁻) and 49F^{ENZr} (PSA⁺) cells with respect to (w.r.t.) 16D^{CRPC} cells derived from the RNA-seq data of Bishop et al²⁷.

Comment 9: The authors state that downregulation of SOX2 using siRNA did not have any effect on the expression of SPINK1, whereas overexpression of SOX2 increases SPINK1 expression. This appears to be a contradictory result, if we are to assume that SOX2 controls SPINK1 expression. Please explain or otherwise address this discrepancy.

Response: We sincerely thank the reviewers for highlighting the discrepancy associated with SOX2-mediated *SPINK1* regulation. Previously, we have done *SOX2* silencing experiment with single transfection of siRNA against *SOX2* (due to limited availability of siRNA) in 22RV1 cells, which resulted in only ~50% reduction in the *SOX2*, and no change in the *SPINK1* expression was found. In the revised manuscript, we performed double transfection of si*SOX2* on 2 consecutive days, and about ~80% reduction in *SOX2* levels with ~50% reduction in *SPINK1* levels was observed (**Revised Fig. 6f and Supplementary Fig. 8c**). Similarly, a significant reduction in the *SPINK1* levels was noted upon *SOX2* knockdown in the androgen-deprived LNCaP-AI cells (**Fig. 6e**), thus confirming *SOX2*-mediated positive transcriptional regulation of *SPINK1*.

Minor concerns:

Comment 10: The discussion section should provide references for statements about SPINK1 expression in prostate cancers.

Response: We have included the references regarding the *SPINK1* expression in several prostate cancer cohorts in the *Discussion* section.

References

1. Tomlins SA, *et al.* The role of SPINK1 in ETS rearrangement-negative prostate cancers. *Cancer cell* **13**, 519-528 (2008).
2. Ateeq B, *et al.* Therapeutic targeting of SPINK1-positive prostate cancer. *Science translational medicine* **3**, 72ra17 (2011).
3. Bhatia V, *et al.* Epigenetic Silencing of miRNA-338-5p and miRNA-421 Drives SPINK1-Positive Prostate Cancer. *Clinical cancer research : an official journal of the American Association for Cancer Research* **25**, 2755-2768 (2019).
4. Tiwari R, *et al.* SPINK1 promotes colorectal cancer progression by downregulating Metallothioneins expression. *Oncogenesis* **4**, e162 (2015).
5. Ogata N. Demonstration of pancreatic secretory trypsin inhibitor in serum-free culture medium conditioned by the human pancreatic carcinoma cell line CAPAN-1. *J Biol Chem* **263**, 13427-13431 (1988).
6. Chen F, *et al.* Targeting SPINK1 in the damaged tumour microenvironment alleviates therapeutic resistance. *Nature Communications* **9**, 4315 (2018).
7. Zhang Y, *et al.* Model-based analysis of ChIP-Seq (MACS). *Genome Biol* **9**, R137 (2008).
8. Phatnani HP, Greenleaf AL. Phosphorylation and functions of the RNA polymerase II CTD. *Genes Dev* **20**, 2922-2936 (2006).
9. Hsin JP, Manley JL. The RNA polymerase II CTD coordinates transcription and RNA processing. *Genes Dev* **26**, 2119-2137 (2012).
10. Bernstein BE, *et al.* A bivalent chromatin structure marks key developmental genes in embryonic stem cells. *Cell* **125**, 315-326 (2006).
11. Zhou VW, Goren A, Bernstein BE. Charting histone modifications and the functional organization of mammalian genomes. *Nat Rev Genet* **12**, 7-18 (2011).
12. Ozaki N, *et al.* Serine protease inhibitor Kazal type 1 promotes proliferation of pancreatic cancer cells through the epidermal growth factor receptor. *Molecular cancer research : MCR* **7**, 1572-1581 (2009).
13. Rasanen K, Itkonen O, Koistinen H, Stenman UH. Emerging Roles of SPINK1 in Cancer. *Clinical chemistry* **62**, 449-457 (2016).
14. Soon WW, *et al.* Combined genomic and phenotype screening reveals secretory factor SPINK1 as an invasion and survival factor associated with patient prognosis in breast cancer. *EMBO molecular medicine* **3**, 451-464 (2011).
15. Hazra A. Using the confidence interval confidently. *J Thorac Dis* **9**, 4125-4130 (2017).
16. Dontu G, *et al.* In vitro propagation and transcriptional profiling of human mammary stem/progenitor cells. *Genes & development* **17**, 1253-1270 (2003).
17. Rajasekhar VK, Studer L, Gerald W, Socci ND, Scher HI. Tumour-initiating stem-like cells in human prostate cancer exhibit increased NF- κ B signalling. *Nature Communications* **2**, 162 (2011).
18. Liu C, *et al.* The microRNA miR-34a inhibits prostate cancer stem cells and metastasis by directly repressing CD44. *Nat Med* **17**, 211-215 (2011).
19. Qin J, *et al.* The PSA(-/lo) prostate cancer cell population harbors self-renewing long-term tumor-propagating cells that resist castration. *Cell Stem Cell* **10**, 556-569 (2012).
20. Shukla S, *et al.* Aberrant Activation of a Gastrointestinal Transcriptional Circuit in Prostate Cancer Mediates Castration Resistance. *Cancer cell* **32**, 792-806 e797 (2017).

21. Olsen JR, *et al.* Context dependent regulatory patterns of the androgen receptor and androgen receptor target genes. *BMC Cancer* **16**, 377 (2016).
22. Marchiani S, *et al.* Androgen-responsive and -unresponsive prostate cancer cell lines respond differently to stimuli inducing neuroendocrine differentiation. *Int J Androl* **33**, 784-793 (2010).
23. Svensson C, *et al.* REST mediates androgen receptor actions on gene repression and predicts early recurrence of prostate cancer. *Nucleic acids research* **42**, 999-1015 (2014).
24. Shen R, Dorai T, Szaboles M, Katz AE, Olsson CA, Buttyan R. Transdifferentiation of cultured human prostate cancer cells to a neuroendocrine cell phenotype in a hormone-depleted medium. In: *Urologic Oncology: Seminars and Original Investigations* (ed[^](eds). Elsevier (1997).
25. Knuutila M, *et al.* Antiandrogens Reduce Intratumoral Androgen Concentrations and Induce Androgen Receptor Expression in Castration-Resistant Prostate Cancer Xenografts. *Am J Pathol* **188**, 216-228 (2018).
26. Padmanaban V, *et al.* E-cadherin is required for metastasis in multiple models of breast cancer. *Nature*, (2019).
27. Bishop JL, *et al.* The Master Neural Transcription Factor BRN2 Is an Androgen Receptor-Suppressed Driver of Neuroendocrine Differentiation in Prostate Cancer. *Cancer Discov* **7**, 54-71 (2017).
28. Koide H, *et al.* Comparison of ERG and SPINK1 expression among incidental and metastatic prostate cancer in Japanese men. *Prostate* **79**, 3-8 (2019).
29. Leinonen KA, *et al.* Association of SPINK1 expression and TMPRSS2:ERG fusion with prognosis in endocrine-treated prostate cancer. *Clinical cancer research : an official journal of the American Association for Cancer Research* **16**, 2845-2851 (2010).
30. Alshalalfa M, *et al.* Characterization of transcriptomic signature of primary prostate cancer analogous to prostatic small cell neuroendocrine carcinoma. *International journal of cancer*, (2019).
31. Beltran H, *et al.* Divergent clonal evolution of castration-resistant neuroendocrine prostate cancer. *Nature medicine* **22**, 298 (2016).
32. Beltran H, *et al.* Whole-Exome Sequencing of Metastatic Cancer and Biomarkers of Treatment Response. *JAMA Oncol* **1**, 466-474 (2015).
33. Beltran H, *et al.* A Phase II Trial of the Aurora Kinase A Inhibitor Alisertib for Patients with Castration-resistant and Neuroendocrine Prostate Cancer: Efficacy and Biomarkers. *Clinical cancer research : an official journal of the American Association for Cancer Research* **25**, 43-51 (2019).
34. Celia-Terrassa T, *et al.* Epithelial-mesenchymal transition can suppress major attributes of human epithelial tumor-initiating cells. *J Clin Invest* **122**, 1849-1868 (2012).
35. Berx G, *et al.* E-cadherin is a tumour/invasion suppressor gene mutated in human lobular breast cancers. *EMBO J* **14**, 6107-6115 (1995).
36. Onder TT, Gupta PB, Mani SA, Yang J, Lander ES, Weinberg RA. Loss of E-cadherin promotes metastasis via multiple downstream transcriptional pathways. *Cancer research* **68**, 3645-3654 (2008).
37. Frixen UH, *et al.* E-cadherin-mediated cell-cell adhesion prevents invasiveness of human carcinoma cells. *J Cell Biol* **113**, 173-185 (1991).

38. Guilford P. E-cadherin downregulation in cancer: fuel on the fire? *Mol Med Today* **5**, 172-177 (1999).
39. He Y, *et al.* Androgen receptor splice variants bind to constitutively open chromatin and promote abiraterone-resistant growth of prostate cancer. *Nucleic acids research* **46**, 1895-1911 (2018).
40. Lu F, Lamontagne J, Sun A, Pinkerton M, Block T, Lu X. Role of the inflammatory protein serine protease inhibitor Kazal in preventing cytolytic granule granzyme A-mediated apoptosis. *Immunology* **134**, 398-408 (2011).

Reviewers' Comments:

Reviewer #1:

Remarks to the Author:

In the revised version of the manuscript "Androgen deprivation upregulates SPINK1 expression and potentiates cellular plasticity in prostate cancer" by Tiwari et al., the authors addressed the major concerns raised by this referee. Nevertheless, few points remain to be addressed in full. To facilitate analysis of the data, the authors should improve the labelling of the Western blots and include size markers.

The Western blots shown in Fig. 2a, d and in Supplementary Fig. 2b are not very informative! Since the size markers are missing, it is impossible to evaluate whether the depicted signals correspond to SPINK1. To prove that the detected band is correct the authors must include a positive and a negative control. In addition, the authors must show by RNAi that the signals correspond to SPINK1. In Fig. 3a and 3b, the SPINK1 gene is represented with opposite orientations. To facilitate understanding, please use the same gene orientation in both pictures.

Reviewer #2:

Remarks to the Author:

In the revised manuscript, the authors revised previous misleading statements and added necessary information to improve the manuscript. They carefully answered the questions raised by the reviewers in a point-to-point manner.

I have the following concerns for the revised manuscript.

The immunoblotting results are confusing.

1. In Fig.4a, the knockdown efficiency of SPINK1 in 22RV1 cells is not optimal (50% efficiency), but its impact on Ecad and Vim expression is so profound, almost comparable to the data in Fig.5I, in which the SPINK1 is very well knocked down (almost 100% efficiency). How do the authors explain the discrepancy?
2. In Fig. 5d, the LNCaP-AI (day 0) seems to be SPINK1-"negative", however, in other panels, LNCaP-AI express adequate amount of SPINK1. At least it appears to be enough to perform knockdown assays.

Reviewer #3:

Remarks to the Author:

Most concerns from the previous have been addressed satisfactorily. I have no further critiques.

Responses to Reviewers' comments:

Reviewer #1 (Remarks to the Author):

In the revised version of the manuscript “Androgen deprivation upregulates SPINK1 expression and potentiates cellular plasticity in prostate cancer” by Tiwari et al., the authors addressed the major concerns raised by this referee. Nevertheless, few points remain to be addressed in full.

Response: We thank Reviewer #1 for his/her time and efforts in going through the revised manuscript and giving positive assessment of the revised manuscript. Points raised by the reviewer #1 has been now addressed and additional supporting data is provided.

Comment 1: To facilitate analysis of the data, the authors should improve the labelling of the Western blots and include size markers. The Western blots shown in Fig. 2a, d and in Supplementary Fig. 2b are not very informative! Since the size markers are missing, it is impossible to evaluate whether the depicted signals correspond to SPINK1.

Response: We apologize to the reviewer for not properly labelling and including protein size markers in the figures depicting Western blots. As suggested, in the revised manuscript, we have now labelled the size (kDa) for the respective proteins for all the immunoblots throughout the manuscript. Furthermore, we are also providing the uncropped immunoblots as the source data for all the Western blot experiments.

Comment 2: To proof that the detected band is correct the authors must include a positive and a negative control. In addition, the authors must show by RNAi that the signals correspond to SPINK1.

Response: We understand reviewer’s concern regarding the correct size of the SPINK1 protein in the immunoblots detected by SPINK1 antibody (R&D Systems, Cat No. MAB7496). As suggested, we re-confirmed the specificity of the SPINK1 antibody for detecting human SPINK1 protein by using appropriate controls. Immunoblot experiment was performed using positive control for SPINK1 (protein lysate of colorectal cancer WiDr cells, endogenously positive for SPINK1), negative controls (lysates from SPINK1-negative benign immortalized prostate epithelial RWPE-1 and prostate cancer PC3 cells). As shown below in **Figure 1**, SPINK1 protein signal in the positive control (WiDr) indeed corresponds to the SPINK1 signal obtained in the stable 22RV1-shSCRM (scrambled control) and RNA interference (RNAi)-mediated *SPINK1* silenced cells (22RV1-shSPINK1-1 and 22RV1-shSPINK1-2).

Please note that previous studies have already reported higher levels of SPINK1 in colorectal cancer cell lines such as WiDr¹, HT-29² and COLO205³. Therefore, we used WiDr as a positive control in the immunoblot experiment.

Figure 1: Western blot showing SPINK1 protein levels using anti-SPINK1 antibody (R&D Systems, Cat No. MAB7496), in WiDr (SPINK1-positive control), 22RV1-shSCRM, 22RV1-shSPINK1-1, 22RV1-shSPINK1-2, and SPINK1-negative controls, RWPE-1 and PC3 cells.

We would also like to mention that same SPINK1 antibody has been previously used in the study published by Yu Chen's group (*Cancer Cell. 2017*)⁴. In this study, the authors used the above mentioned antibody to detect the expression of SPINK1 protein in 22RV1 cells (**Figure 2**), known to be endogenously positive for SPINK1 expression^{4, 5, 6}.

Figure 2: Western blot showing SPINK1 protein levels using anti-SPINK1 antibody (R&D Systems, Cat No. MAB7496), in vehicle or doxycycline treated derivatives of 22RV1 cells⁴ (Figure adapted from Shukla et al., *Cancer Cell. 2017*).

Comment 3: In Fig. 3a and 3b, the SPINK1 gene is represented with opposite orientations. To facilitate understanding, please use the same gene orientation in both pictures.

Response: We appreciate the reviewer's observation about the orientation of the *Fig. 3a and 3b*. As suggested, in the revised figure we have changed the orientation of ARE sites in the schema of *SPINK1* promoter similar to the ChIP-Seq data (*Revised Fig. 3a, lower panel*).

Reviewer #2 (Remarks to the Author):

In the revised manuscript, the authors revised previous misleading statements and added necessary information to improve the manuscript. They carefully answered the questions raised by the reviewers in a point-to-point manner.

Response: We thank the Reviewer #2 for his/her positive assessment of the revised manuscript. We have also addressed the concerns raised him/her in the following section of this letter.

Comment 1: The immunoblotting results are confusing. In Fig.4a, the knockdown efficiency of SPINK1 in 22RV1 cells is not optimal (50% efficiency), but its impact on E-cad and Vim expression is so profound, almost comparable to the data in Fig.5I, in which the SPINK1 is very well knocked down (almost 100% efficiency). How do the authors explain the discrepancy?

Response: We understand the reviewer's concern regarding the knockdown efficiency of SPINK1 in 22RV1 (*Fig. 4a*) as compared to LNCaP-AI cells (*Fig. 5I*). Please note that previous immunoblot for 22RV1 cells with shRNA-mediated *SPINK1* knockdown (*Fig. 4a*) was captured at slightly higher exposure, hence the knockdown efficiency, which is ~70-80% was not evident. Now, in the new immunoblot shown below, we have taken lower (1 minute) and higher (2 minutes) exposures of the immunoblots for SPINK1. Please note that at lower exposure the knockdown efficiency is around ~70-80% as compared to scramble control (**Figure 3** in the response letter), hence the change observed in the E-Cad and Vimentin expression is profound. This new immunoblot data for SPINK1-knockdown in 22RV1 cells has also been incorporated in the revised *Fig. 4a* of the manuscript.

Figure 3: Western blot showing SPINK1 protein levels at lower and higher exposure times in 22RV1-shSCRM, 22RV1-shSPINK1-1 and 22RV1-shSPINK1-2 cells.

The knockdown efficiency of a shRNA depends upon the stoichiometry between shRNA and its respective target mRNA⁷. Moreover, 22RV1 cells harbor abundant endogenous SPINK1 levels, while wildtype LNCaP cells are SPINK1-negative in nature (**Supplementary Fig. 2a, b**), and only express *SPINK1* transcript when subjected to long-term androgen deprivation (denoted as LNCaP-AI cells) (**Fig. 5c-d**). Since *SPINK1* transcript is abundant in 22RV1 cells, therefore higher efficiency of *SPINK1* knockdown was not achieved. Conversely, LNCaP-AI cells upon androgen deprivation start expressing lower levels of SPINK1 (even at day 10 and 20). As shown in schema (**Fig. 5j**), wildtype LNCaP cells were used to generate stable LNCaP-shSPINK1 and LNCaP-shSCRM cells, and subsequently subjected to androgen deprivation for 30 days. We speculate that LNCaP-shSPINK1 cells are already synthesizing abundant shRNA against *SPINK1*, hence as soon as SPINK1 transcript is generated due to androgen deprivation, it's getting degraded due to the effect of RNA interference, thus eliciting higher knockdown efficiency in LNCaP-AI (shSPINK1-1 and shSPINK1-2) cells (**Fig. 5l**).

Comment 2: In Fig. 5d, the LNCaP-AI (day 0) seems to be SPINK1- “negative”, however, in other panels, LNCaP-AI express adequate amount of SPINK1. At least it appears to be enough to perform knockdown assays.

Response: We understand reviewer’s concern and apologise for not properly explaining the experimental conditions of the LNCaP-AI cells. The LNCaP cells are inherently SPINK1-negative (as shown in **Supplementary Fig. 2a, b**), and only start expressing SPINK1 when subjected to long-term androgen deprivation (**Fig. 5d**). In **Fig. 5d**, the LNCaP-AI at day 0 are in fact wildtype LNCaP cells, which were not subjected to androgen deprivation, hence are negative for SPINK1 expression (**Fig. 5d**), whereas LNCaP-AI cells upon androgen deprivation (at day 10, 20 and 30) show elevated levels of SPINK1.

As depicted in the schema (**Fig. 5j**), in this particular experiment, we stably transfected wildtype LNCaP cells with shRNA against *SPINK1* and scrambled control shRNA to generate stable LNCaP-shSPINK1 and LNCaP-shSCRM cells respectively. Subsequently, LNCaP-shSPINK1 and LNCaP-shSCRM cells were subjected to long term androgen deprivation (denoted as LNCaP-AI-shSPINK1 and LNCaP-AI-shSCRM), and were assessed for the SPINK1 expression at day 30. As shown in **Fig. 5l**, the LNCaP-AI-shSCRM cells upon androgen deprivation for 30 days, expressed high level of SPINK1 protein, however, in LNCaP-AI-shSPINK1, due to the presence of shRNA against *SPINK1* transcript, showed a significant reduction in the SPINK1 levels due to RNA interference. To avoid any confusion, in the revised **Fig. 5k-m**, we have now labelled the LNCaP-AI cells as “LNCaP-AI (30 days)”,

for indicating the androgen deprivation period of 30 days in case of LNCaP-AI-shSCRM and LNCaP-AI-shSPINK1 cells.

Reviewer #3 (Remarks to the Author):

Comment: Most concerns from the previous have been addressed satisfactorily. I have no further critiques.

Response: We thank Reviewer #3 for his/her time and efforts in evaluating the revised manuscript. We also acknowledge his/her constructive suggestions, which improved the overall quality of the manuscript.

References:

1. Tiwari R, *et al.* SPINK1 promotes colorectal cancer progression by downregulating Metallothioneins expression. *Oncogenesis* **4**, e162 (2015).
2. Gouyer V, *et al.* Autocrine induction of invasion and metastasis by tumor-associated trypsin inhibitor in human colon cancer cells. *Oncogene* **27**, 4024-4033 (2008).
3. Ida S, *et al.* SPINK1 Status in Colorectal Cancer, Impact on Proliferation, and Role in Colitis-Associated Cancer. *Molecular cancer research : MCR* **13**, 1130-1138 (2015).
4. Shukla S, *et al.* Aberrant Activation of a Gastrointestinal Transcriptional Circuit in Prostate Cancer Mediates Castration Resistance. *Cancer Cell* **32**, 792-806 e797 (2017).
5. Tomlins SA, *et al.* The role of SPINK1 in ETS rearrangement-negative prostate cancers. *Cancer cell* **13**, 519-528 (2008).
6. Ateeq B, *et al.* Therapeutic targeting of SPINK1-positive prostate cancer. *Science translational medicine* **3**, 72ra17 (2011).
7. Cuccato G, Polynikis A, Siciliano V, Graziano M, di Bernardo M, di Bernardo D. Modeling RNA interference in mammalian cells. *BMC systems biology* **5**, 19 (2011).

Reviewers' Comments:

Reviewer #2:

Remarks to the Author:

The authors largely answered my questions. I have no further comments.